# Interactions between *Epichloë* endophyte and the plant microbiome impact nitrogen responses in host *Achnatherum inebrians* plants

Yuanyuan Jin,[1,2,3,4,5,6] Zhenjiang Chen,[1,2,3,4,5,6] James F. White,[7] Kamran Malik,[1,2,3,4,5,6] Chunjie Li[1,2,3,4,5,6]

**ABSTRACT** The clavicipitaceous fungus *Epichloë gansuensis* forms symbiotic associations with drunken horse grass (*Achnatherum inebrians*), providing biotic and abiotic stress protection to its host. However, it is unclear how *E. gansuensis* affects the assembly of host plant-associated bacterial communities after ammonium nitrogen ($NH_4^+$-N) treatment. We examined the shoot- and root-associated bacterial microbiota and root metabolites of *A. inebrians* when infected (I) or uninfected (F) with *E. gansuensis* endophyte. The results showed more pronounced $NH_4^+$-N-induced microbial and metabolic changes in the endophyte-infected plants compared to the endophyte-free plants. *E. gansuensis* significantly altered bacterial community composition and β-diversity in shoots and roots and increased bacterial α-diversity under $NH_4^+$-N treatment. The relative abundance of 117 and 157 root metabolites significantly changed with *E. gansuensis* infection under water and $NH_4^+$-N treatment compared to endophyte-free plants. Root bacterial community composition was significantly related to the abundance of the top 30 metabolites [variable importance in the projection (VIP) > 2 and VIP > 3] contributing to differences between I and F plants, especially alkaloids. The correlation network between root microbiome and metabolites was complex. Microorganisms in the Proteobacteria and Firmicutes phyla were significantly associated with the R00693 metabolic reaction of cysteine and methionine metabolism. Co-metabolism network analysis revealed common metabolites between host plants and microorganisms.

**IMPORTANCE** Our results suggest that the effect of endophyte infection is sensitive to nitrogen availability. Endophyte symbiosis altered the composition of shoot and root bacterial communities, increasing bacterial diversity. There was also a change in the class and relative abundance of metabolites. We found a complex co-occurrence network between root microorganisms and metabolites, with some metabolites shared between the host plant and its microbiome. The precise ecological function of the metabolites produced in response to endophyte infection remains unknown. However, some of these compounds may facilitate plant-microbe symbiosis by increasing the uptake of beneficial soil bacteria into plant tissues. Overall, these findings advance our understanding of the interactions between the microbiome, metabolome, and endophyte symbiosis in grasses. The results provide critical insight into the mechanisms by which the plant microbiome responds to nutrient stress in the presence of fungal endophytes.

**KEYWORDS** *Achnatherum inebrians*, *Epichloë gansuensis*, alkaloids, root exudates, root plant specialized metabolism, microbiome, bacterial microbial communities

The perennial grass *Achnatherum inebrians* is highly adaptable, allowing it to dominate degraded grasslands across northwest China from 1,700 to 4,300 m

Address correspondence to Zhenjiang Chen, chenzhenjiang@lzu.edu.cn, or Chunjie Li, chunjie@lzu.edu.cn.

The authors declare no conflict of interest.

See the funding table on p. 15.

elevation. This 30-million-hectare range makes it an important species for controlling soil erosion and stabilizing sandy soils (1). However, its common symbiosis with the fungal endophytes *Epichloë gansuensis* and *A. inebrians* poses problems for livestock. These endophytes increase the plant's resilience, productivity, and adaptability but produce toxic alkaloids (ergine and ergonovine) that can poison grazing animals (2, 3). Replacing infected grasses with endophyte-free cultivars is impractical because the fungi provide ecological benefits (4, 5).

The advantages of the *Epichloë* symbiosis are well-studied. The fungi enhance the grass's drought and cold tolerance (6, 7), growth (8, 9), nutrient uptake (10–12), disease and pest resistance (13–16), tolerance of heavy metals (17, 18), and protecting the biodiversity of over-grazed grasslands (19). They induce physiological and morphological changes in the host such as altered root-shoot ratios (8), increased photosynthesis (9), and shifts in sugars, amino acids, and enzymes (18). While the benefits of *Epichloë* endophyte to host health and growth are clear, the influence on above- and below-ground microorganisms is less understood.

*E. gansuensis* infection altered the composition and diversity of the bacterial and fungal communities in phyllosphere and seed of *A. inebrians* in the field (20, 21), but there have been surprisingly few effects of environmental components that interact with *Epichloë* endophyte on plant microbiome. Comparative studies reveal endophyte-related differences in root-soil archaea and bacteria involved in nitrogen cycling (22). The fungi also increase root fungal diversity under drought and alter soil bacterial communities (23–25). However, the mechanisms behind these microbiome shifts are unknown.

The symbiosis alters host metabolism by changing pathways and alkaloid production (26, 27). These changes affect litter decomposition and soil microflora (28–30). Foliar endophytes likely influence root microbes through both litter chemistry and root exudates (31). Nevertheless, we lack a comprehensive understanding of the specialized metabolites produced, their functions, and their microbial effects (32, 33). Among the unknowns are the dynamics of microorganisms from host shoot and root tissues during growth, the metabolic processes that are engaged during microbial changes, and regulatory relationships between root microbes and root exudates. Knowledge of the latter is important if we hope to clarify the effect of foliar endophytes on leaf, root, and soil microorganisms in drunken horse grass via integrating microbiology and metabolomics analysis.

This knowledge gap motivates the current study. We will monitor changes in shoot and root microbes and exudates in *A. inebrians* with and without *E. gansuensis* under different nitrogen regimes. The goals are to identify responsive microorganisms and metabolites and elucidate how endophyte symbiosis affects plant-microbe interactions and specialized metabolism under varying nutrient availability. Overall, this introduction summarizes the current state of knowledge, identifies key gaps, and outlines an approach to advance our understanding of the complex interplay between the microbiome, metabolome, and symbiosis in this ecologically important grass.

## MATERIALS AND METHODS

### Plant material and treatment

*E. gansuensis* endophyte-infected (I) and -free (F) *A. inebrians* seeds were collected from E+ and E− seed fields and planted at Yuzhong Experimental Station of Lanzhou University, Gansu Province, China, in 2017. All the seeds were stored at 4°C in the dark until the following experiments. I and F seeds were planted in a 72-hole plate of cultivation seedlings filled with vermiculite (2.0 kg), which was sterilized at 150°C for 24 h. After 45 days, endophyte status detection in seedlings was performed by staining and microscopic examination following the method of Welty (34). All the I seedlings infected with endophyte and then eight seedlings that differed only in the presence or absence of the *Epichloë* endophyte were selected for the following experiments, respectively. Endophyte-infected seedlings and endophyte-free seedlings

were transferred into water (W) and 0.1 mmol/L ammonia-N (A) nutrient solutions. Plants were grown in a greenhouse at 25°C ± 2°C with a 16 h:8 h, light:dark photoperiod (c. 600 µmol.m$^{-2}$. s$^{-1}$) in a random block design and at 8 wk post-cultivation. Root exudates were collected using a root exudates collection device after plants reached 7 days of growth. Leaf and root tissues of I and F plants ($n = 4$ per group) under different nutrient treatments were collected after 9 wk (eight wk + 7 days) at a consistent time and immediately flash-frozen in liquid nitrogen (N$_2$).

## Ergonvine extraction and determination

Alkaloid was extracted in two stages using a two-phase solvent system following a method modified by Miles et al. (35). In brief, the seedlings were frozen in the refrigerator at −20°C for 5 h and then dried in a freeze dryer at −60°C for 24 h. The freeze-dried material was ground to a fine powder with a mortar. A 50 mg powder sample was taken into 2 mL centrifuge tube, and 1 mL 20% ice acetic acid was added, followed by ultrasonication for 5 min, shaking for 2 min, and centrifugation for 5 min at 1,000 r/min. The alkaloids extract was collected with 1 mL miscible liquids of 95% methanol and 5% ammonia after purification by filtration through a OmniPac PCX-100 HPLC Columns. The PCX columns were activated with 2 mL of methanol before use. The collected extract (0.25 mL) was filtered into a 1.5 mL brown chromatography vial through a 0.22 µm pore size membrane filter. The samples in the bottles were stored in the refrigerator at 4°C under dark conditions awaiting high-performance liquid chromatography (HPLC) detection.

Ergonvine was quantified on an Agilent 1100 HPLC system, with a 250 × 4.6 mm Eclipse XDB-C18 column containing 5 µm particles. Mobile phase: 0.1M NH4OAc (A): 0.1M CH$_3$CN = 3:1, at a flow rate of 1.0 mL/min, 5 µL injection volume, and open the wash needle (wash with 10% isodiol). The eluent gradients: 0.0–10.0 min: 95% A: 5% B; 10.0–25.0 min: 75% A: 25% B; 35.0–50 min: 50% A: 50% B; 51.0–55.0 min: 5% A: 95% B; 56.0–71.0 min: 95% A: 5% B. The detection wavelength (Ex) of a fluorescence detector was set at 312 nm, the emission wave (Em) was 427 nm, and the column temperature was 25°C. Ergonvine was purchased from Sigma-Aldrich China. The 0 mg/L, 0.375 mg/L, 0.75 mg/L, 1.5 mg/L, 3 mg/L, 6 mg/L, and 12 mg/L concentrations of ergometrine standard solutions were configured by diluting the mother liquor of the standard (20 mg/L). The relevant standard linear equations were established by the external standard method: $Y_{Ergonovine} = 154.98X − 10.152$, $R^2 = 0.9981$, Y: peak areas, and X: concentration (mg/L). The process was monitored, and the peak area was determined using the Chem Station for LC Rev.A.10.01, USA. Alkaloid levels in injection volume were identified and quantified by comparing their peak areas with external standard curves. Ergonvine contents in plant samples were calculated according to the relevant equation and the dilution of the sample.

## DNA extraction, PCR amplification, and sequencing

Total DNA was extracted from 30 mg of leaves or roots of endophyte-infected and endophyte-free plants ($n = 4$) either nitrogen-treated or well-watered (control) using a modified cetyltrimethylammonium bromide (CTAB) DNA extraction procedure (36). The concentration and purity of extracted DNA were determined using the Micro Nanodrop ND-1000 UV-Vis Spectrophotometer (Nanodrop Technologies, Wilmington, DE, USA), and DNA extract quality was checked on 1% agarose gel. DNA amplification was done using an ABI GeneAmp 9700 and TransStart Fastpfu DNA Polymerase (TransGen AP221-02) with primer sequences (338F: 5′-ACTCCTACGGGAGGCAGCAG-3′ and 806R: 5′-GGACTACHVGGGTWTCTAAT-3′) (37) and with fluorescence quantification using Quantus Fluorometer (Promega, USA). Four of the six biological replicates with the highest DNA quality were selected for 16S rRNA sequencing. Preparation of PCR product libraries and PCR amplification and sequencing was performed at Majorbio Bio-Pharm Technology Co., Ltd. (Shanghai, China). In brief, following PCR amplification analysis and quantitation, purified PCR product libraries were generated using a NEBNext

Rapid DNA-Seq Kit and sequenced on Illumina MiSeq PE300 (Illumina Inc., San Diego, CA, USA) sequencing platform generating 20–60 million 413 bp paired-end reads per sample. The 16S rRNA nucleotide sequence analysis was performed on the free online platform of Majorbio Cloud (www.majorbio.com) using the QIIME-1.9.1 pipeline (38) (http://qiime.sourceforge.net/). Briefly, low-quality reads (length <50 bp or with a quality value <20 or having N bases) were removed by fastp (39) (https://github.com/OpenGene/fastp, version 0.19.6). Filtered, high-quality PE reads were spliced according to overlap relationship using flash (http://www.cbcb.umd.edu/software/flash, version 1.2.11) and were clustered for operational taxonomic unit (OTU) based on 97% similarity using uparse (40) (http://drive5.com/uparse/, version 7.1), and chimeras were removed. Species taxonomic annotation with a confidence threshold of 70% was based on best matches to databases from Silva 16S rRNA gene (v138) using Ribosomal Database Project (RDP) classifier (http://rdp.cme.msu.edu/, version 2.11).

## Metabolite extraction

Here, metabolites were extracted from 100 µL liquid sample with 400 µL methanol: acetonitrile [1:1 (vol/vol)] solution by sonicating at 40 kHz for 30 min at 5°C. Samples were placed at −20°C for 30 min to precipitate proteins. The resultant samples were centrifuged for 15 min (13,000 $g$, 4°C) to remove solid particles. The supernatant was carefully transferred to new microtubes and evaporated to dry under a gentle stream of nitrogen. After desiccation, samples were reconstituted in 100 µL loading solution of acetonitrile:water [1:1 (vol/vol)] by sonicating briefly in a 5°C water bath. Extracted metabolites were centrifuged for 15 min (13,000 $g$, 4°C), and cleared supernatants were transferred into fresh vials and stored at −80°C prior to liquid chromatography-mass spectrometry (LC-MS) analysis.

## UHPLC-MS/MS analysis

Ultra-HPLC-tandem mass-spectrometry (UHPLC-MS/MS) analysis was performed in positive and negative mode (3,500 V to −3,500 V) following Li et al. (41) established protocol. In brief, metabolite analyses were performed on a Thermo UHPLC -Q Exactive HF-X Mass Spectrometer equipped with electrospray ionization (ESI) source operating (Thermo, Waltham, Massachusetts, USA). Metabolite separation (2 µL sample) was conducted on an HSS $T_3$ column (100 × 2.1 mm, 1.8 µm; Waters) using 0.1% formic acid in water:acetonitrile [95:5 (vol/vol); solvent A] and 0.1% formic acid in acetonitrile:isopropanol:water [47.5:47.5:5 (vol/vol); solvent B] as mobile phase and the following parameters: flow rate of 0.4 mL min$^{-1}$; column temperature 40°C; 2 µL injection; method: 0–3.5 min (0%–24.5% B; 0.4 mL.min$^{-1}$), 3.5–5–5.5 min (24.5%, 65% to 100% B; 0.4 mL.min$^{-1}$), 5.5–7.4 min (100% B; 0.4–0.6 mL.min$^{-1}$), 7.4–7.6–7.8 min (100%, 51.5% to 0% B; 0.6, 0.6 to 0.5 mL.min$^{-1}$) and 7.8–9–10 min (0% B, 0.5–0.4 and 0.4 mL.min$^{-1}$). The optimal parameters: heater temperature of 425°C; capillary temperature of 325°C; sheath gas flow rate at 50 arb; aux gas flow rate at 13 arb; ion-spray voltage floating. Mass spectra were acquired in continuum mode over m/z 70–1,050 using data-dependent MS/MS acquisition (DDA), with normalized collision potential scanned between 20, 40, and 60 V rolling for DDA. Full MS resolution was 60,000, and MS/MS resolution was 7,500. Quality control samples (QC) and reference samples were analyzed every 10 injections in order to monitor the stability of the LC-MS system.

## Data processing and annotation

After the mass spectrometry detection was completed, the obtained DDA data were processed using PROGENESIS QI (v.3.0; Waters, Milford, MA, USA) using line baseline filterion, retention time (RT) alignment, mass correction, peak recognition and integration (filtering low-quality peaks), adduct grouping and deconvolution, and the following parameters: peak picking with the automatic sensitivity method: default settings; RT range: 0.1–8.0 min. Annotated metabolites were defined by RT and m/z information

(designated as features). Human metabolome database (HMDB, http://www.hmdb.ca/), Kyoto Encyclopedia of Genes and Genomes (KEGG, http://www.genome.jp/kegg/), Metlin (https://metlin.scripps.edu/), and Majorbio databases were used to provide feature annotations based on 10 ppm precursor mass tolerance, 95% isotope similarity, and 10 ppm fragment mass tolerance. Metabolic features detected at least 80% in any set of samples were retained. Before downstream statistical analyses, ion abundances of each of the metabolic features were normalized to the internal standard telmisartan by sum, based on four biological replicates. Meanwhile, variables with relative standard deviation (RSD) >30% of QC samples were removed, and log10 logarithmization was performed to obtain the final data matrix.

## Differential metabolite mining

The normalized and transformed data were used for statistical analysis using the free online platform of majorbio cloud (42) (cloud.majorbio.com). Briefly, orthogonal partial least squares discriminant analysis (OPLS-DA) was conducted using ROPLS package (Version 1.6.2) in R, and afterward, the stability of the model using 7-cycle interactive validation was evaluated. Differentially expressed metabolites were identified based on the variable importance in the projection (VIP) obtained by the OPLS-DA model, the $P$-value of Student's $t$ test, fold change (FC) and the following parameters: VIP > 1; $P < 0.05$; FC < 1 or FC > 1, default. Significantly different metabolites between endophyte-infected and endophyte-free plants under different N treatment were summarized and mapped into their biochemical pathways through metabolic enrichment and pathway analysis using the KEGG database. In addition, SCIPY.STATS (Python packages, https://docs.scipy.org/doc/scipy/) and Fisher's exact test ($P < 0.05$) were used to identify statistically significantly enriched pathways. Different metabolites were classified based on the pathways in which they are produced or the functions they perform.

## Statistical analyses

OTUs of all samples were classified as abundant (relative abundances > 0.1%), intermediate (0.01% < relative abundances < 0.1%), and rare (relative abundances < 0.01%) based on their relative abundances (43). All the statistical analyses were conducted in R version 3.3.1 (http://www.r-project.org) and heat maps of community composition were created using the VEGAN package. The differences in the diversity index of alpha were tested by analysis of variance (ANOVA) followed by the independent sample $t$ test. The similarity of bacterial community structure in the underground and aboveground in *A. inebrians* plants was examined using principal coordinate analysis (PCoA) based on the bray-curtis distance algorithm. Differential microbial community structure was identified based on the results in analysis of similarities (ANOSIMs) and permutational multivariate analysis of non-parametric variance tests. Significantly different abundances in bacterial taxa between endophyte-infected and endophyte-free plants were identified based on adjusted $P$-value ($P_{adj}$) <0.05 and Wilcoxon rank-sum test as selection criteria at the phylum level. Pearson's correlations between bacterial taxa and metabolites were calculated using the STATS package in R. The correlation between bacterial community structure and metabolite abundances was tested by partial Mantel tests using the VEGAN package.

## Sankey network analysis

We used Bio-Sankey network and STA-Sankey network analysis in MetOrigin software to mine the correlation between bacteria at different classification levels (i.e., at phylum, class, order, family, and genus) and metabolites in a metabolic pathway (44). The left censored data quantile regression interpolation (QRILC) and percentage calculation methods were used to interpolate the missing values and normalize the data before the statistical analysis (45). The OTUs correlation matrix was calculated using Spearman's rank-based correlation coefficient (Spearman's $r$). Correlations were considered statistically robust when Spearman's $r > 0.65$ and $P < 0.05$. We analyzed potential bacteria

that might participate in a metabolic reaction by the Bio‐Sankey network. Metabolite, metabolic reaction, and associated bacteria at different taxonomic levels were connected by bands gray as background. Data of statistically significant correlations between bacteria and a metabolite would be analyzed and highlighted in red and green color, indicating positive and negative correlations. Statistical correlation between bacteria and a metabolic reaction was explored using the STA‐Sankey network. They would be highlighted in red or green color if there exists a biological relationship between bacteria and metabolic reaction by MetOrigin database searching. STA-Sankey was complementary to Bio-Sankey results and focuses on statistical correlations between bacteria and metabolites in real data sets, which may explain the potential relationships between them. The Bio-Sankey and STA‐Sankey networks could visualize the biological and statistical correlations between bacteria and root exudates straightforwardly.

## RESULTS

### Differential assemblage of shoot- and root-associated bacterial microbiota in response to the presence or absence of *Epichloë* endophyte

The findings of community composition analysis revealed that N addition had the greatest significant impact on the change of the plant microbiome (Fig. S1; Fig. 1). Furthermore, it was found that shoot microbiomes in shoots of endophyte-infected plants consisted mostly of Cyanobacteria, Proteobacteria, Bacteroidota, and Actinobacteriota under nitrogen treatment, accounting for >90% of total relative abundance, with Cyanobacteria dominating the shoot-associated bacteria of endophyte-free plants (Fig. 1a). The root-associated bacterial communities of endophyte-infected plants, at the phylum level, were mostly dominated by Proteobacteria (41.71%), Bacteroidota (37.62%), Cyanobacteria (9.16%), and Actinobacteriota (5.64%), but for endophyte-free plants, Firmicutes (65.34%) and Proteobacteria (24.71%) were the most dominant phyla (Fig. 1b). It was observed that the *E. gansuensis* infection significantly increased the relative abundance of Proteobacteria in roots from 24.63% to 42.18% relative to endophyte-free plants, Bacteroidota from 4.72% to 37.47%, and Actinobacteriota from 1.27% to 5.63% (Fig. 1c through f). The rrn copy number of rare communities was significantly changed by the presence of the endophyte under A-N treatment (Fig. S2).

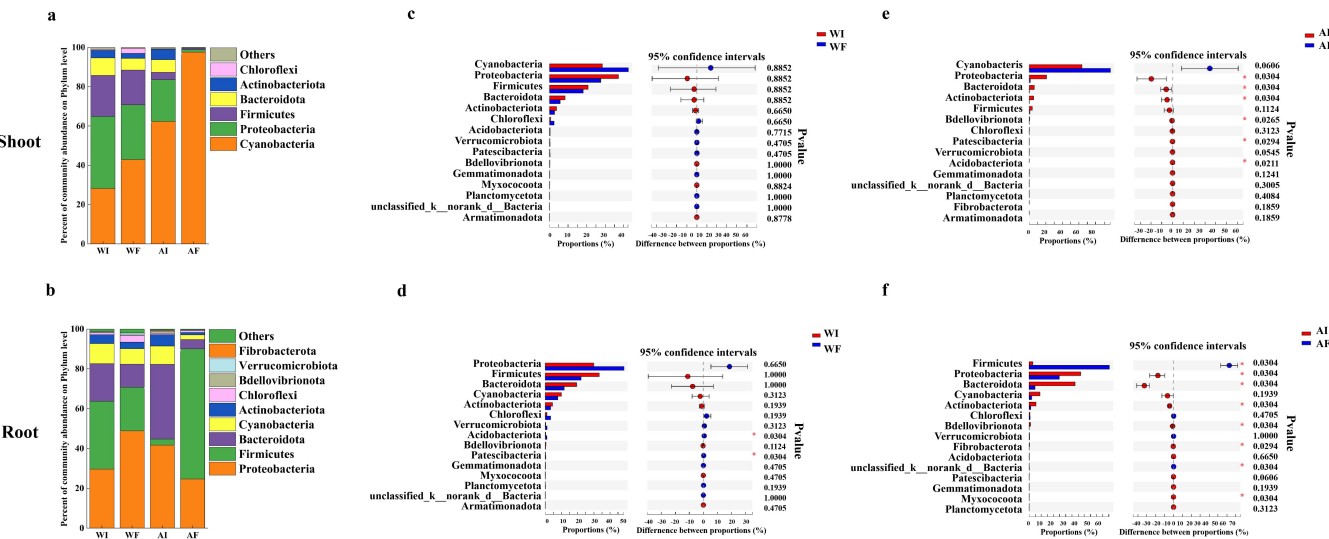

**FIG 1**   (a–b) Bar plots showing the composition of the relatively abundant bacterial genera in shoot (a) and root (b) affected by endophyte infection under water (W) and ammonia-N (A) treatments. (c–f) Wilcoxon rank-sum test bar plots showing differential microbial communities analysis of the relatively abundant bacterial phyla in shoot (c and e) and root (d and f) between endophyte-infected plants (I) and endophyte-free plants (F) under water (W; c and d) and ammonia-N (A; e and f) treatments. X-axis represents bacterial microbes at different phylum levels. Different colored boxes indicate endophyte-infected plants (I) and endophyte-free plants (F). Y-axis represents the average relative abundance of bacterial phyla between I and F plants.

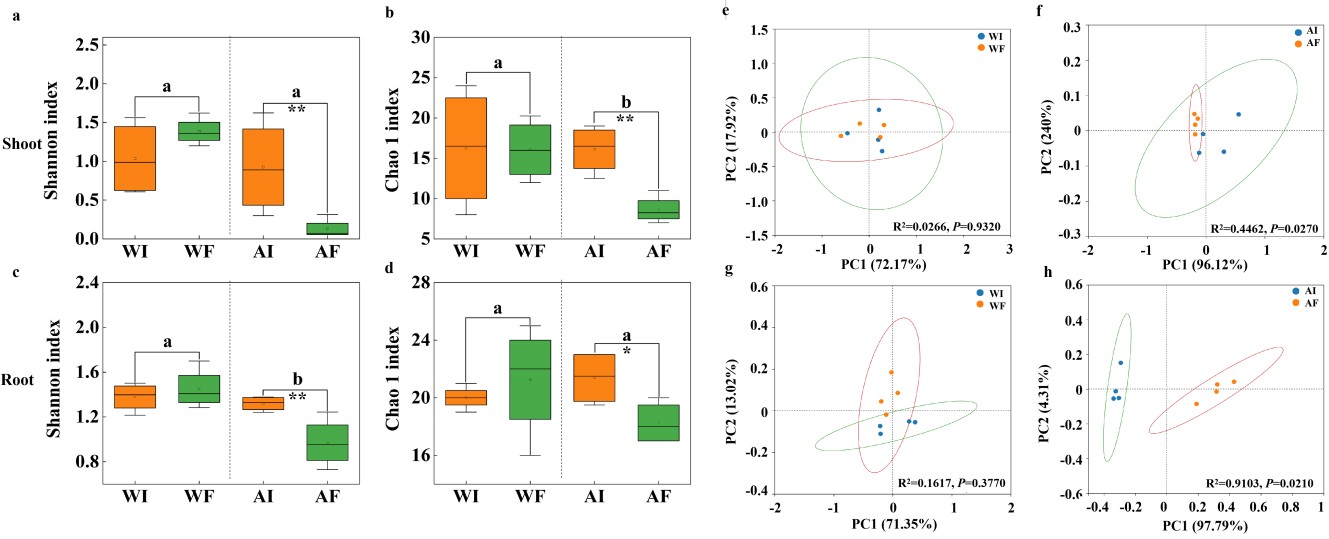

**FIG 2** (a–d) The independent sample *t* test for the effects of endophyte infection on alpha diversity of bacterial phyla in shoot (a and b) and root (c and d) under water (W) and ammonia-N (A) treatments. a and c: Shannon index; b and d: Chao 1 index. (e–h) PCoA showing similarity or difference in shoot (e and f) and root (g and h) of bacterial community composition between endophyte-infected plants (I) and endophyte-free plants (F) under water (W; e and g) and ammonia-N (A; f and h) treatments.

The Shannon index and Chao 1 index of bacterial communities in shoots and roots were significantly ($P < 0.05$) influenced by the presence of the *Epichloë* endophyte under N treatment but showed no significant difference under water treatment (Fig. 2a through d). Meanwhile, PCoA was used to visualize the β-diversity based on the Bray-Curtis dissimilarity, and ANOSIM analyses were conducted to evaluate the significance of changes in community structure ($P < 0.05$) between endophyte-infected plants and endophyte-free plants (Fig. 2e through h).

## *A. inebrians-E. gansuense* symbiont shows N-inducible root metabolite alterations

We identified 3,003 and 2,991 mass spectrometry peaks in positive and negative ion mode (QC) samples (Table S1). In total, 604 and 207 metabolites were successfully detected from I and F roots of *A. inebrians* in positive and negative (POS and NEG) modes before data preprocessing, respectively. To eliminate or reduce the error caused by the analysis process, 580 and 198 positive and negative (POS and NEG) modes differential metabolites with names features were selected for downstream statistical analysis after filtering of low-quality peaks, missing value padding, data normalization, and RSD assessment of QC (Table S1). Nitrogen treatment altered the ratio of the down- and up-regulated metabolites in root exudates in POS and NGE modes between endophyte-infected plants and endophyte-free plants (Fig. 3a and b; Fig. S3). The OPLS-DA model results showed that these detected metabolites were significantly diverse between I and F root under A-N (R2X = 0.564, R2Y = 0.992, and Q2Y = 0.855) and water (R2X = 0.858, R2Y = 0.999, and Q2Y = 0.924) treatments (Fig. 3c and d).

Aligned with the microbiomic changes, the analysis revealed that *E. gansuense*-infected *A. inebrians* had a greater number of specific metabolites than *E. gansuense*-free *A. inebrians* under N treatment (Fig. 4b). These differential metabolites could be divided into nine classes, including organic acids and derivatives, lipids and lipid-like molecules, organo-heterocyclic compounds, organic oxygen compounds, alkaloids and derivatives, and Benzenoids (Supplemental File 2; Table S2). Our analysis of the top 30 metabolites (VIP > 2 and VIP > 3) contributing to the difference between I and F plants revealed that most compounds significantly increased in abundance by endophyte infection under A-N treatment (Fig. 4c), whereas a significant decrease was observed under water

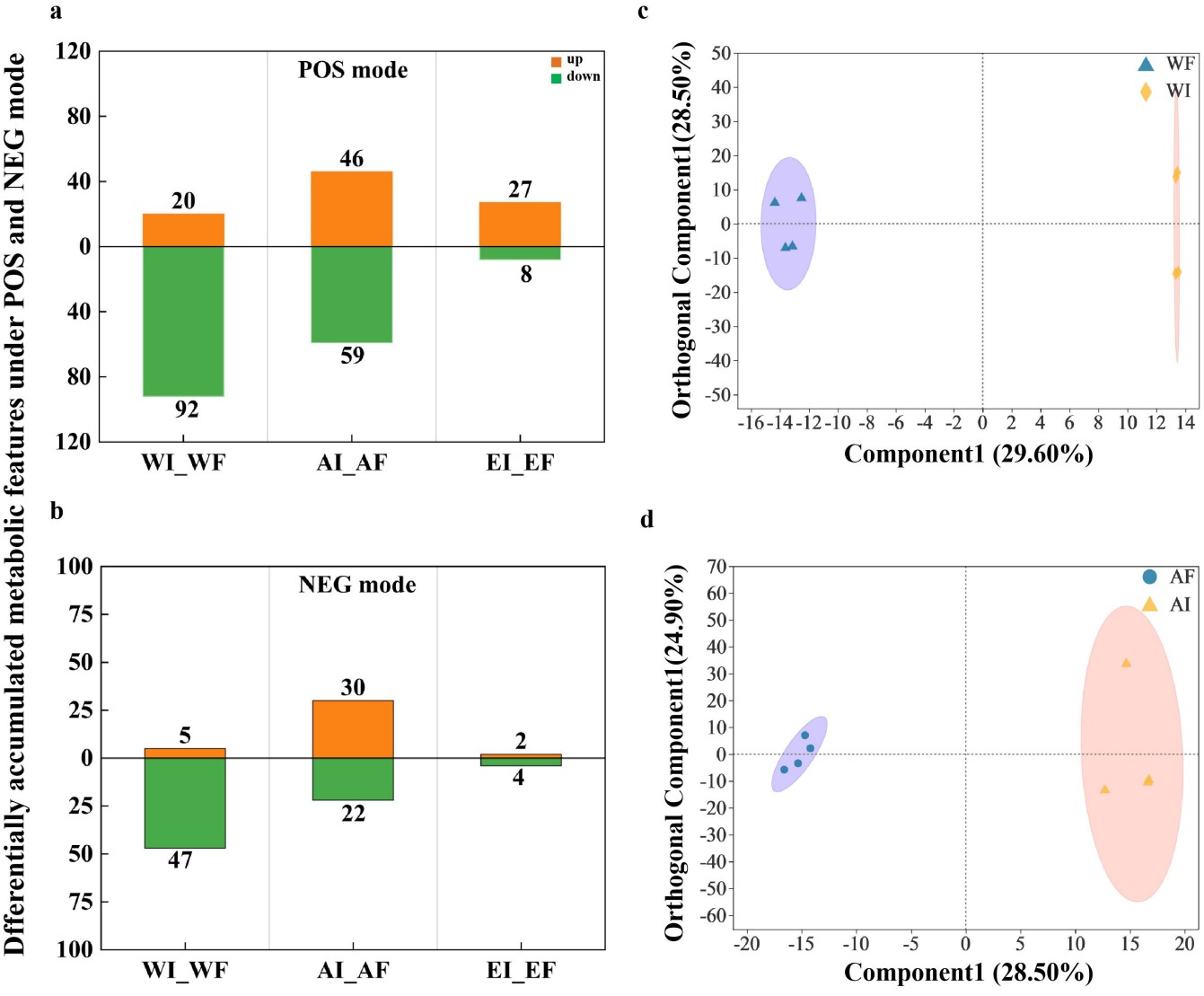

**FIG 3** (a and b) Metabolic compounds that showed significant differences in root exudates of endophyte-infected plants (I) and endophyte-free plants (F) under water (W) and ammonia-N (A) treatments. (a and b) Number of differentially accumulated metabolic features that were detected under POS (a) and NEG (b) modes, respectively; orange and green bars indicate the numbers of up-regulated and down-regulated metabolic features in relative abundance. (c and d) OPLS-DA plots of UHPLC-MS/MS positive mode metabolome divergence between endophyte-infected plants (I) and endophyte-free plants (F) under water (W; c) and ammonia-N (A; d) treatments. X-axis: first predicted principal component explanatory degree; Y-axis: first orthogonal component explanatory degree.

treatment compared to endophyte-free plants root (Fig. 4d). Among the up-annotated signature metabolites detected under water treatment, general classification metabolites were highly enriched (Fig. 4c). In contrast, except for the significant enrichment of general metabolites, some specific metabolites were also significantly enriched in the root of endophyte-infected plants root under A-N treatment (Fig. 4d; Supplemental File 2). The specific metabolites identified by HMDB searches as 6-allyl-8b-Carboxy-ergo-line, Cinchonidine, Norajmaline, and (2-Naphthalenyloxy) acetic acid were significantly accumulated in roots of endophyte-infected plants compared to endophyte-free plants (abundance all zero) under A-N treatment (Fig. 4e, f, h and i). In addition to ergometrine, which was significantly increased by 732-fold in roots of endophyte-infected plants than roots of endophyte-free plants (Fig. 4g), the results of the $t$-test showed that ergonovine content in shoot under water ($P = 0.005$) and A-N ($P < 0.001$) treatments were significantly increased by endophyte infection (Fig. S4).

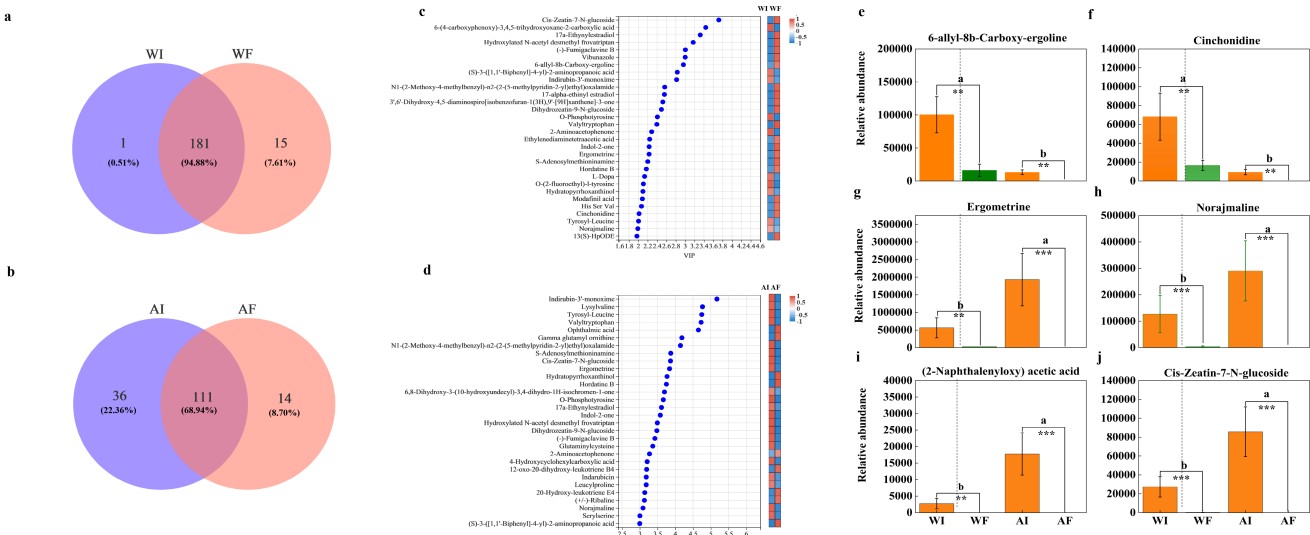

**FIG 4** Venn plot showing the number of common and unique metabolites in root exudates of endophyte-infected plants (I) and endophyte-free plants (F) under water (W; a) and ammonia-N (A; b) treatments. (c and d) VIP bar plot and clustered heat-map showing metabolite expression patterns and VIP of metabolites in multivariate statistical analysis in root exudates of endophyte-infected plants (I) and endophyte-free plants (F) under water (W; c) and ammonia-N (A; d) treatments. VIP_value: the contribution value of the metabolite to the difference between I and F plants. Left: metabolite VIP bubble plots. Y-axis: metabolites. X-axis: VIP value. The metabolites were arranged according to the size of the VIP value, from top to bottom. Right: metabolite expression heat-map. Heat-map columns: I and F plants. The red/blue color indicates the magnitude of the relative expression of the metabolites in root exudates of I and F plants, and the correspondence between the color gradient and the value is shown in the gradient color block. (e–j) Special differentially expressed metabolites between endophyte-infected plants (I) and endophyte-free plants (F). (e) 6-Allyl-8b-Carboxy-ergoline. (f) Cinchonidine. (g) Ergometrine. (h) Norajmaline. (i) (2-Naphthalenyloxy) acetic acid. (j) Cis-Zeatin-7-N-glucoside.

There were six metabolite pathways significantly enriched ($P < 0.05$) in *A. inebrians* under water treatment (Fig. S5a). Compared to water treatment, the significantly enriched pathways for 157 metabolic compounds in I and F plants by the hypergeometric distribution algorithm and relative-betweenness centrality were phenylalanine metabolism ($P = 0.002$), aminoacyl-tRNA biosynthesis ($P = 0.030$), lysine degradation ($P = 0.025$), tryptophan metabolism ($P = 0.035$), and cysteine and methionine metabolism ($P = 0.036$) under A-N treatment (Fig. S5b).

## Visualization of biological and statistical correlation between bacterial community and root exudates using Sankey network

According to the metabolites from microbiota, three significant co-metabolism-associated metabolic pathways were identified, associated with nutrient supply, including cysteine and methionine metabolism, tyrosine metabolism, and arginine and proline metabolism. For instance, five differential metabolites (i.e., S-adenosylmethioninamine, spermidine synthase, adenosylmethionine decarboxylase, S-adenosyl-L-methionine, and 5′-methylthioadenosine) were involved in the cysteine and methionine metabolic pathway, which participated in three different metabolic reactions (R00178, R01920, and R02869). Proteobacteria, Firmicutes, Actinobacteria, and Bacteroidetes phyla were closely associated with metabolic reaction R00693 in the Bio-Sankey network (Fig. 5a). The relative abundance of S-Adenosylmethioninamine in root metabolism was significantly up-regulated in endophyte-infected plants than endophyte-free plants under A-N treatment (Supplemental File 2). The relative abundance of *Bacillus*, *Paenibacillus*, and *Exiguobacterium* genera in roots was the most significantly increased in *A. inebrians* plants under A-N treatment (dark red if $P < 0.05$) and was higher in endophyte-free plants compared to endophyte-infected plants, which were closely associated with metabolic reaction R01920 in parallel (Fig. 5a). Statistical correlation analysis confirmed

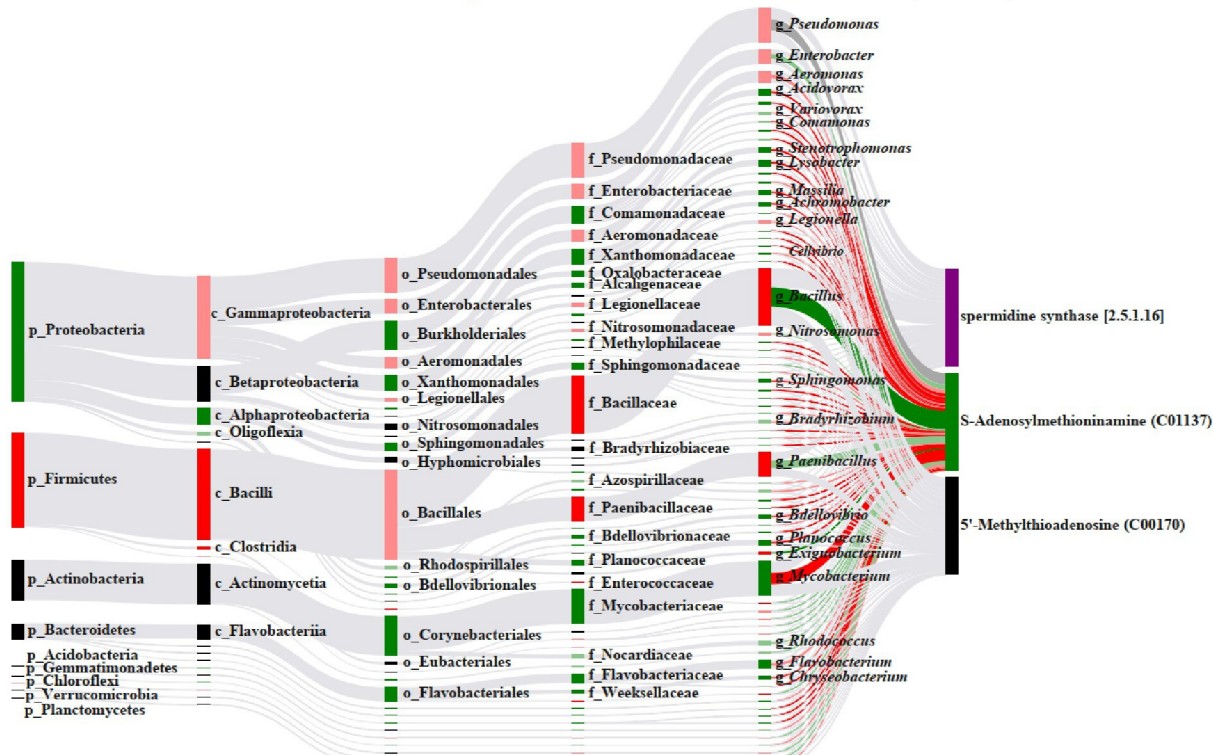

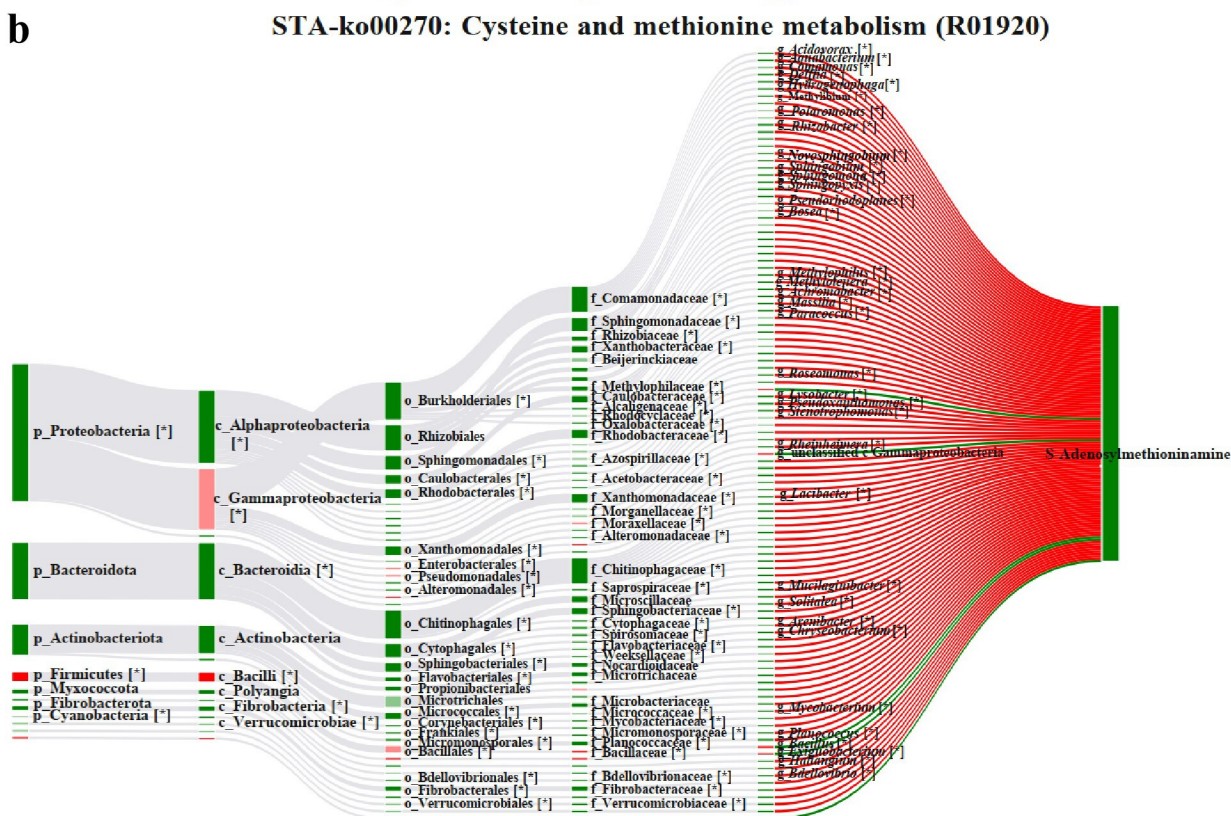

FIG 5   (a) The BIO-Sankey Network for R01920 metabolic reaction in cysteine and methionine metabolism. (b) The STA-Sankey Network for R01920 metabolic reaction in cysteine and methionine metabolism. *: bacteria with statistically significant correlations with metabolites. Red/green nodes: up-/down-regulation bacteria or metabolites. Red/green bands: the positive/negative correlations with metabolites. Dark red/green color: the statistically significance $P < 0.05$. The widths of the bands between nodes were linearly proportional to the number of bacteria; therefore, the wider band indicated the deeper involvement of a group of bacteria.

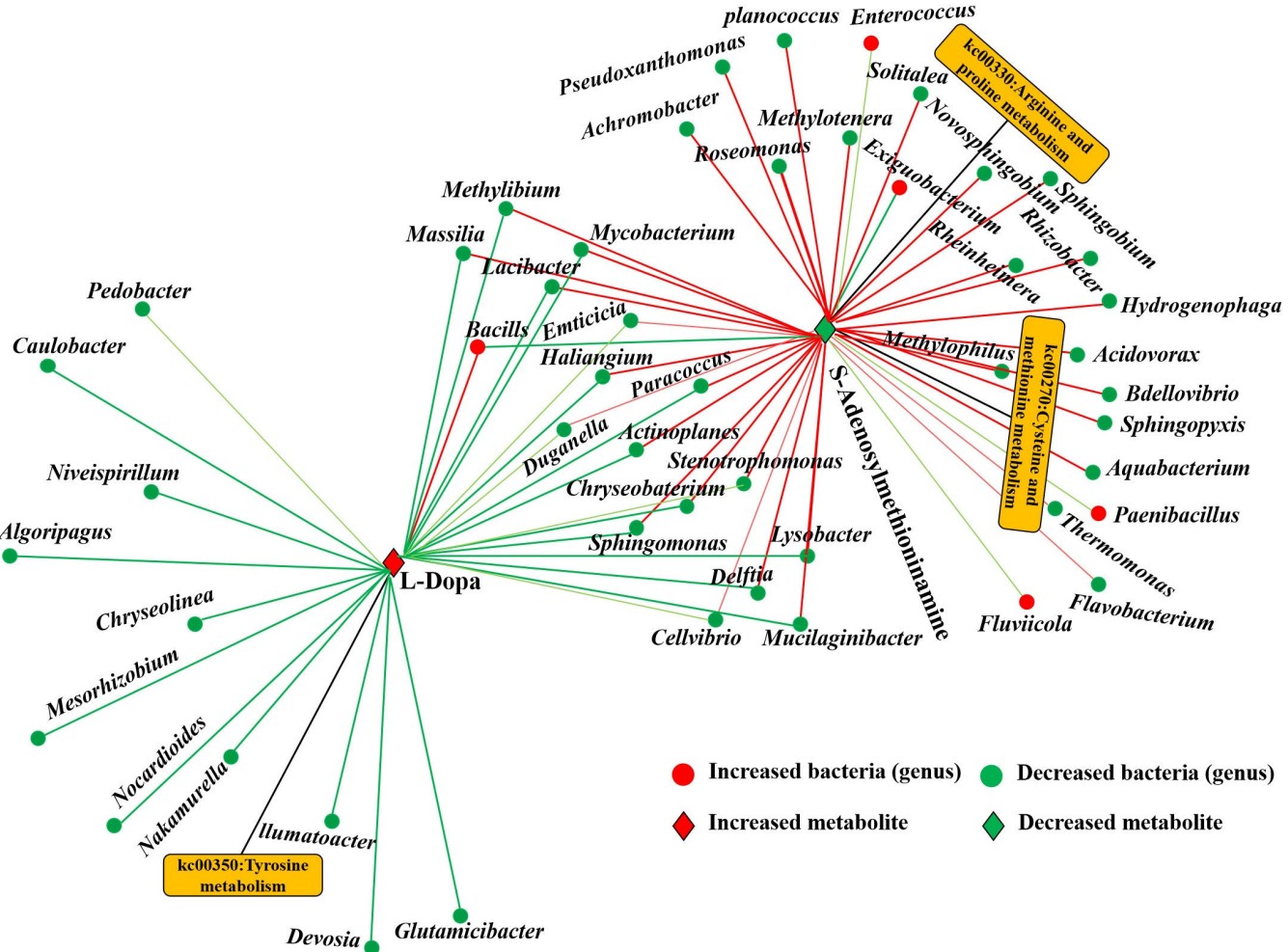

**FIG 6** Co-metabolism network shows differential metabolites shared by the host and microbiota and their related microbes. Diamond and dot shapes: metabolites and microbes, correspondingly. Rectangular frames: metabolic pathways. Red/green nodes: up-/down-regulation bacteria or metabolites. Red/green lines: the positive/negative correlations between microbes and metabolites.

that Proteobacteria phylum was closely associated with S-adenosylmethioninamine in the metabolic reaction R00693 by the STA-Sankey network (Fig. 5b).

## Co-metabolism network shows differential metabolites shared by the host and microbiota and their related microbes

The co-metabolism network of cysteine and methionine metabolism, tyrosine metabolism, and arginine and proline metabolism showed that two root metabolites were associated with 49 differential root bacteria under A-N treatment ($P < 0.05$; Fig. 6). With the exception of very few bacteria, for example, *Enterococcus*, *Exiguobacterium*, *Paenibacillus*, and *Fluviicola* genera, most of bacterial communities were significantly and positively correlated with S-adenosylmethioninamine. However, we were able to find a significant positive effect of L-dopa metabolites only on bacteria of the genus *Bacills* but significant negative correlation with other bacterial communities (Fig. 6). Bacteria-root metabolite association network provided more specific information on metabolic changes associated with *A. inebrians-E. gansuense* symbiont plants under A-N treatment.

## DISCUSSION

By performing biological and statistical correlation analyses between root bacteria and metabolites, this study provides novel evidence for the numerous close and intricate interrelationships between the two factors. The metabolomic data showed that the *E. gansuensis* endophyte altered metabolic components in host plants. Integration of microbiome and metabolome data sheds light on the complex bidirectional interactions between endophytes, host plants, and microbial communities. Our multi-omic characterization of endophyte-infected *A. inebrians* advances understanding of the intricate network of signals mediating tripartite symbiosis. Further elucidation of the specific mechanisms underlying microbiome-metabolome interplay will be key. Overall, this systems biology approach coupling amplicon sequencing and metabolomics enables greater insight into the multidimensional impacts of endophytes on host-microbe interactions.

### Effect of the *E. gansuensis* endophyte on the composition of plant bacterial microbial communities

The current study found that *A. inebrians* plants infected with the *E. gansuensis* endophyte harbored more complex phyllosphere microbial communities compared to uninfected plants (Fig. 1a and b). These results align with previous studies by Liu et al. and Liang et al. showing altered microbiome composition and diversity in the phyllosphere, seeds, and roots of endophyte-infected *A. inebrians* under optimal conditions (20, 21). Extensive research on other *Epichloë* endophytes like *Epichloë occultans*, *Epichloë bromicola*, and *Epichloë festucae* var. *lolii* has also shown that root and rhizosphere microbiomes exhibit complex responses to endophyte symbiosis (46–48). Our findings further demonstrate that shifts in plant microbial communities can be driven by the presence of fungal endophytes (22, 25), reflecting the critical role endophytes play in host adaptation.

The current study demonstrated that the A-N treatment induced significant alterations in bacterial community composition in both shoots and roots of endophyte-infected plants compared to endophyte-free plants (Fig. 1a and b). This manifested as a greater fold-change increase in bacterial abundance in infected vs uninfected plants under A-N conditions (Fig. 1e and f). These findings illustrate an amplified nitrogen-responsive effect of the endophyte, aligning with previous research showing that the influence of *Epichloë* endophytes on host plants is enhanced in nitrogen-rich environments (49). As proposed by prior studies, copiotrophic bacteria thrive in nutrient-rich conditions, while oligotrophic bacteria with efficient nutrient utilization prevail in nutrient-poor settings (50). Accordingly, we found endophyte-infected shoots harbored more rare bacteria compared to endophyte-free shoots under A-N, while the opposite trend occurred in roots, mirroring measured nitrate and ammonium levels (Fig. S6). As rare microbes play key roles governing nitrogen availability and plant nutrient uptake (51), the endophyte-mediated shifts in low-abundance bacteria may have significant ecological impacts.

Numerous studies demonstrate that alpha diversity of shoot and root microbiomes critically regulates plant health, growth, and environmental adaptability (52, 53). Our work revealed the *Epichloë* endophyte markedly increased alpha and beta diversity of shoot and root bacteria under A-N but not under water-only conditions (Fig. 2), aligning with previous findings (20, 21, 54). Taken together, these results suggest that nitrogen availability enables endophytes to substantially augment the diversity and complexity of host-associated bacterial communities. Further investigation of the multifaceted interplay between nitrogen, endophytes, and microbiota will provide greater insight into the factors shaping holobiont resilience. Our integrative analysis of amplicon data with plant metabolomic and nutrient profiles represents a valuable systems biology approach to elucidating the intricate tripartite network of endophyte-host-microbe interactions.

## Effect of the *E. gansuensis* endophyte on root metabolites

This study demonstrated that *E. gansuensis* infection was the primary driver of metabolic profile alterations in *A. inebrians*, as endophyte-infected and endophyte-free plants showed distinct profiles even under optimal conditions (21). Notably, we observed amplified nitrogen-responsive metabolic divergence between infected and uninfected plants (Fig. 3), aligning with previous findings on low-nitrogen responses (49). General and specialized metabolism pathways exhibited pronounced differences between endophyte-infected and endophyte-free roots under A-N (Fig. 4), mirroring the accumulation of various organic acids, amino acids, and fatty acids reported by Hou et al. (49). Other metabolomic studies reveal *Epichloë* endophytes can modulate host stress resilience, growth, and soil insect distributions by altering root exudate composition and metabolic pathways (55, 56).

We found that *A. inebrians* alkaloid contents increased under salt and drought stresses, particularly cytotoxic ergonovine (57). Similarly, under A-N conditions, endophyte-infected plants showed significant ergometrine accumulation in root exudates (Fig. S4), contrasting related stress-induced alkaloids like ergine and ergonovine (49, 58). The major contribution of toxic pyrrolopyrazine and ergot alkaloids to nitrogen uptake and translocation in infected plant shoots may relate to the concurrent accumulation of carbohydrates and amino acids, as seen in endophyte-infected tall fescue (58).

Overall, our dual metabolomic and amplicon sequencing analyses provide novel insight into the close interrelationship between the endophyte-conferred alkaloid profile and the structure of the root microbiome. The significant correlations found between specific bacterial taxa and ergot alkaloids hint at a complex codependency. Further elucidation of the factors regulating alkaloid biosynthesis and secretion, the bioactivities of these compounds against different microbes, and subsequent microbial community shifts will be key. This holistic multi-omic approach enables system-level understanding of the multifaceted chemical signals mediating dynamic endophyte-host-microbe interactions belowground.

## Root microbial and metabolic interactions with *A. inebrians-E. gansuense* symbiont

Prior studies show root exudates are the primary nutrient source for rhizosphere microbes, directly shaping ecological and compositional dynamics (59). Notably, correlation analysis revealed significant positive associations between root metabolite levels and bacterial taxa, suggesting endophyte infection may recruit communities by modulating metabolites (Fig. S7). Microbes chemotactically locate root tips using exuded sugars, organic acids, and amino acids but are then degraded by reactive oxygen species as the plant extracts nutrients from colonizers (60, 61). We found that the antioxidant glutamylcysteine was markedly accumulated in endophyte-infected roots under A-N, potentially mitigating reactive oxygen and improving nutrient extraction from bacteria.

The co-occurrence network demonstrated significant interrelationships between the root microbiome and metabolome in A. inebrians (Fig. 5). For instance, we observed nitrogen-induced accumulation of S-adenosylmethioninamine exclusively in endophyte-infected roots, highlighting metabolite co-metabolism between bacteria and the symbiotic holobiont (44).

Overall, the integration of amplicon and metabolomic data provides greater insight into the chemical dialog regulating tripartite interactions belowground. Our findings reveal that endophytes modify root exudate profiles to selectively enhance colonization by beneficial microbiota, highlighting the key role of plant-microbe signaling in symbiotic assembly and function. Further characterization of metabolite bioactivities, specialized microbial nutritive requirements, chemoattraction dynamics, and community interdependencies will elucidate the multifaceted mechanisms enabling robust, mutually beneficial endophyte-host-bacteria relationships.

## Conclusions

Nitrogen significantly amplified the effects of the *Epichloë* endophyte on host-associated microbes and metabolites compared to water-only conditions. Our results demonstrate the *E. gansuense* endophyte-induced substantial alterations in shoot and root bacterial community composition and beta diversity under $NH_4^+$-N treatment. Relative abundance of rare taxa in shoots and alpha diversity in both shoots and roots was higher in infected vs uninfected plants under nitrogen fertilization. Concurrently, endophyte infection modulated root metabolite classes and abundances, with variations in alkaloid levels linked to shifts in the root microbiome assembly. Complex and significant correlations were found between root bacteria and metabolites, highlighting their multifaceted interplay. This integrated amplicon sequencing and metabolomics approach provides novel insight into the specialized metabolites secreted by endophyte-infected plants and their relevance in recruiting beneficial microbiota. Our findings exemplify the power of combinatorial multi-omics to elucidate the intricate chemical dialog mediating tripartite endophyte-host-microbe interactions belowground. Further characterization of metabolite bioactivities, chemoattraction mechanisms, and microbial dependencies will be key to deciphering the complex signaling enabling robust, mutually beneficial symbioses. Overall, this systems biology strategy represents a valuable tool for rapidly advancing the discovery of bioactive compounds and clarifying their ecological roles.

### ACKNOWLEDGMENTS

We thank the editor and anonymous reviewers for their valuable comments.

Financial support for this work was provided by the National Basic Research Program of China (2014CB138702), the National Science Foundation of China (32201445), the China Postdoctoral Science Foundation (2021M701525), Program for Changjiang Scholars and Innovative Research Team in University, China (IRT17R50), Gansu Provincial Youth Science and Technology Fund Program (22JR5RA532), Gansu Province Outstanding Doctoral Students Project (22JR5RA434), and 111 Project (B12002). The authors are thankful for support from USDA-NIFA Multistate Project W4147 and the New Jersey Agricultural Experiment Station.

Y.J.J., Z.J.C., and C.J.L. conceived the original research. Y.J.J. analyzed data analysis and wrote the manuscript. Y.J.J. conducted plant nitrogen incubation experiments and plant growing, plant harvesting, sampling, and sample processing. Y.J.J. performed microbiomic and metabolomic analysis. J.F.W., K.M., and C.J.L. assisted in the writing of the original article draft and revision of manuscript.

### AUTHOR AFFILIATIONS

[1]State Key Laboratory of Herbage Improvement and Grassland Agro-ecosystems, Lanzhou University, Lanzhou, China

[2]Key Laboratory of Grassland Livestock Industry Innovation, Ministry of Agriculture and Rural Affairs, Lanzhou University, Lanzhou, China

[3]Engineering Research Center of Grassland Industry, Ministry of Education, Lanzhou University, Lanzhou, China

[4]Gansu Tech Innovation Center of Western China Grassland Industry, Lanzhou University, Lanzhou, China

[5]Center for Grassland Microbiome, Lanzhou University, Lanzhou, China

[6]College of Pastoral Agriculture Science and Technology, Lanzhou University, Lanzhou, China

[7]Department of Plant Biology, Rutgers University, New Brunswick, New Jersey, USA

### AUTHOR ORCIDs

Zhenjiang Chen 🔟 http://orcid.org/0000-0002-2833-5668
James F. White 🔟 http://orcid.org/0000-0002-6780-7066

Chunjie Li ⓘ http://orcid.org/0000-0002-3287-2140

## FUNDING

| Funder | Grant(s) | Author(s) |
|---|---|---|
| Intellectual Property Plan (Targeted Organization) Project of Gansu Administration for Market Regulation | 22ZSCQD01 | Chunjie Li |
| The National Science Foundation of China | 32201445 | Zhenjiang Chen |
| The China Postdoctoral Science Foundation | 2021M701525 | Zhenjiang Chen |
| Gansu Province Outstanding Doctoral Students Project | 22JR5RA434 | Chunjie Li |
| Gansu Provincial Youth Science and Technology Fund Program | 22JR5RA532 | Zhenjiang Chen |
| The Fundamental Research Funds for the Central Universities | lzujbky-2022-kb02, lzujbky-2023-49 | Yuanyuan Jin |
| Gansu Province Grassland Monitoring and Evaluation Technology Support Project of Gansu Province Forestry and Grassland Administration | [2021] 794 | Chunjie Li |
| USDA-NIFA Multistate Project | W4147 | James F. White |

## AUTHOR CONTRIBUTIONS

Yuanyuan Jin, Conceptualization, Data curation, Formal analysis, Software, Validation, Writing – original draft, Writing – review and editing | Zhenjiang Chen, Conceptualization, Data curation, Formal analysis, Funding acquisition, Software, Writing – review and editing | James F. White, Writing – review and editing | Kamran Malik, Writing – review and editing | Chunjie Li, Conceptualization, Funding acquisition, Project administration, Resources, Supervision, Writing – review and editing

## DATA AVAILABILITY

The sequence data associated with this project have been deposited in the NCBI (accession number: SUB13084834).

## ADDITIONAL FILES

The following material is available online.

### Supplemental Material

**Tables S1 to S3 and Figure S1 to S7 (Spectrum02574-23-s0001.pdf).** Tables and Figures associated with microorganisms and metabolites of I and F plants.
**Supplemental Excel file (Spectrum02574-23-s0002.xls).** Differential metabolite classification.

### Open Peer Review

**PEER REVIEW HISTORY (review-history.pdf).** An accounting of the reviewer comments and feedback.

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
