## [Reviewer comments · Microbiology Spectrum]

Microbiology Spectrum

Interactions between *Epichloë* endophyte and the plant microbiome impact nitrogen responses in host *Achnatherum inebrians* plants

Yuanyuan Jin, Zhenjiang Chen, James F. White, Jr., Kamran Malik, and Chunjie Li

Corresponding Author(s): Chunjie Li, Lanzhou University

Review Timeline:

Submission Date:	June 20, 2023
Editorial Decision:	November 30, 2023
Revision Received:	January 16, 2024
Accepted:	January 24, 2024

Editor: Victor Gonzalez

Reviewer(s): The reviewers have opted to remain anonymous.

Transaction Report:

DOI: <https://doi.org/10.1128/spectrum.02574-23>

Re: Spectrum02574-23 (**Interactions between *Epichloë* endophyte and the plant microbiome impact nitrogen responses in host *Achnatherum inebrians* plants**)

Dear Prof. Chunjie Li:

Thank you for the privilege of reviewing your work. Below you will find my comments, instructions from the Spectrum editorial office, and the reviewer comments.

Despite the initial interest in the work, there are several issues with the presentation of the results, manuscript editing, and the selection of data to focus on in a single narrative. Although it may seem trivial, effectively communicating the results and conclusions is crucial; otherwise, poor narratives hinder the discovery of the manuscript's message. Therefore, to ensure a fair analysis of the manuscript, I suggest a thorough review of all sections. This includes rewriting the abstract and focusing on both the results and the discussion. The manuscript could benefit from considering only essential figures. In addition to these improvements, it is essential to present a well-formatted text and address the concerns raised by the reviewers.

Revision Guidelines

Sincerely,
Victor Gonzalez
Editor
Microbiology Spectrum

Reviewer #1 (Comments for the Author):

Im sure there are some interesting results in this work but they are difficult to find under the serious problems concerning the English and grammar. Furthermore, there are major flaws to the methodology of the work:

- There is no explanation to how Epichloë was assessed in seedlings or how many seedlings contained the endophyte after assessment. This is a fundamental part of the experiments.
- For the ergonovine determination, a reference (Li et al) is cited for the extraction method but this reference is a poster abstract and contains no valid information whatsoever.
- A standard curve is mentioned for ergonovine but no source or purity of the standard is given.
- For the microbiome analysis the primers that were used are not suitable for plant studies as they are well known to detect large amounts of non-target DNA (e.g. plant chloroplast and mitochondrial DNA). It looks like this mistake was made as the primers were taken from a paper (Liu et al) that investigated bacteria from wastewaters. Therefore, Liu et al did not extract microbial DNA from plants. It is possible to use blockers but the authors never mentioned this. I instead have suggested the use of alternative primers that are more suitable for amplification of DNA from plant material.
- The UHPLC analysis also refers to citations that are not suitable. For instance, the Gonzalez et al 2011 paper is cited for a protocol but this citation is concerned with UHPLC of rat brain samples. The citation contains no protocol matching the current paper and does not contain any instrument settings.

Please see the attached WORD document for a more detailed review.

Reviewer #2 (Comments for the Author):

The manuscript by authors explained the important group of pathogenic endophyte Epichloë and its interaction within the host plants. Most of the endophytes are beneficial for host, however, due to drastic changes in the environment, some of these leads to pathogenic state and results in serious losses to the host. The metabolites from host play significant role in healthy state.

Specific comments:

1. Does the author only focus on the host metabolites ? or the microbial metabolites that are present inside the plants? During treatment not only the host but microbial metabolites are also produced in different response. How could you differentiate plant and microbial metabolites?
2. Line 93. Delete this line "The results of this work are presented here."
3. Please add data of culturable endophytes if you have after you treated the host plant and check the microbial diversity.
4. How about quantification of metabolites ?

Interactions between *Epichloë* endophyte and the plant microbiome impact nitrogen responses in host *Achnatherum inebrians* plants

Yuanyuan Jin,^{a#} Zhenjiang Chen,^{a#} James F. White,^b Kamran Malik,^a Chunjie Li^{a,*}

^aState Key Laboratory of Herbage Improvement and Grassland Agro-ecosystems; Key Laboratory of Grassland Livestock Industry Innovation, Ministry of Agriculture and Rural Affairs; Engineering Research Center of Grassland Industry, Ministry of Education; Gansu Tech Innovation Centre of Western China Grassland Industry; Centre for Grassland Microbiome; College of Pastoral Agriculture Science and Technology, Lanzhou University; Lanzhou 730000, China

^bDepartment of Plant Biology, Rutgers University, New Brunswick, New Jersey 08901, USA

#: Yuanyuan Jin and Zhenjiang Chen contributed equally to this study. The order was decided by seniority.

*Correspondence: Chunjie Li, TEL. +8613919861685, E-mail: chunjie@lzu.edu.cn

ABSTRACT Plants can deploy species-specific, specialized metabolites as mediators of nutrient acquisition, plant defense and modulators of community structure. By contrast, the specialized chemical metabolites produced by *Epichloë* endophytes as mediators of nutrient acquisition, plant defense and modulators of rhizosphere community structure are largely unknown. To investigate specialized metabolic in root exudates of endophyte-infected plants, and bacterial community composition in shoot and root, we integrated metabolome and microbiome analyses of endophyte-infected (I) vs endophyte-free (F) plants under 0.1 mmol/L ammonia-N (A) treatment. *E. gansuensis*-infected and *E. gansuensis*-free *A. inebrians* plants showed significantly more microbiome and metabolome changes under ammonia-N (A) treatment compared to water (W) treatment. *Epichloë* symbiosis significantly changed the bacterial community composition in shoots and roots of *A. inebrians* plants. Endophyte-infected plants had greater Shannon and Chao 1 indexes in shoots and roots, relative abundance of rare bacteria in shoots, and lower rare microbe abundances in roots of endophyte-free plants, when grown in A-N nutrient solutions. Endophyte infection altered root metabolism of *A. inebrians* plants. Metabolomic analyses identified some common and specific metabolic compounds. Root exudates containing (2-Naphthalenyloxy)acetic acid and Cis-Zeatin-7-N-glucoside growth regulator, 6-allyl-8b-Carboxy-ergoline,

cinchonidine and norajmaline alkaloids were produced in only endophyte-infected plants under A-N nutrient treatment, and ergometrine content was up to 732-fold over that of endophyte-free plants. Three significant co-metabolism associated metabolic pathways between host *A. inebrians* and root bacteria were found by Sankey network analysis, that is, cysteine and methionine metabolism, tyrosine metabolism, arginine and proline metabolism. Proteobacteria phylum were closely associated with S-Adenosylmethioninamine in the metabolic reaction R01920 of cysteine and methionine metabolism. It is unknown precisely how plants or fungi use the metabolites produced in response to endophyte infection, however some of these metabolites may somehow relate to plant-microbe symbiosis, and results in increased absorption of bacteria from soil into plant tissues.

IMPORTANCE *Epichloë gansuensis* and *E. inebrians* are endophytic fungi that form symbiotic associations with *Achnatherum inebrians*, valued for its biotic and abiotic stress protection effects, and N deficiency tolerance. The present study was to reveal responses of shoot and root bacterial communities, root exudate composition and both interaction to low nitrogen in foliar *Epichloë* endophyte-infected *A. inebrians* plants by integrating microbiome and metabolomic analysis. *Epichloë* symbiosis significantly increased abundance and diversity of the bacterial community composition in shoots and roots of *A. inebrians* plants. Compared with endophyte-free plants, the relative abundance of most general classification metabolites and specific metabolites in root exudates under A-N nutrient treatment were also significantly increased by *Epichloë* infection. Rhizosphere bacterial community structure was positively and significantly correlated with these metabolites, and there was also three significant co-metabolism associated metabolic pathways between the two.

KEYWORDS *Achnatherum inebrians*, *Epichloë gansuensis*, alkaloids, root exudates, root plant specialized metabolism, microbiome, bacterial microbial communities.

The toxic bunchgrass, drunken horse grass (*Achnatherum inebrians*), is so adaptable to a wide range of environments in subtropical to temperate zones that it dominates over 30 million ha. of degraded grasslands in Northwest China from an elevation of 1700 to 4300 m. Drunken horse grass may be used for soil conservation, wind and sand control in non-grazing areas such as deserts and sandy lands (1). Symbiosis of drunken horse grass with the seed-transmissible endophytic fungi, *Epichloë gansuensis* and *E. inebrians* (2, 3) confers biotic and abiotic stress tolerance and is a significant factor in grass stand persistence, productivity,

and adaptability. However, the most common *E. gansuensis* and *E. inebrians* strains in China, associated with drunken horse grass, produce ergot alkaloids (ergine and ergonovine) that can cause episodes of 'A. *inebrians* toxicosis' to grazing livestock restricting the development of animal husbandry in China (4-6). The use of endophyte-free cultivars is not practically possible due to the high area covered by the grass and the strong competitive and invasive ability of endophyte infected plants of *A. inebrians* (7, 8). *Epichloë* spp. confer several beneficial traits to their grass host including greater resistance to drought stress (9, 10) and low temperature stress (11), improved utilization of soil nitrogen (14, 15), increased phosphorus uptake (16), greater cadmium tolerance (18, 19) and enhanced resistance to diseases (20-22) and pests (23, 24) that can result in increased dry matter (12, 13). . Physiological responses and morphological changes, such as altered crown-root ratio (12), regulated stomatal movement (10), increased photosynthesis (13), and changes in soluble sugar and amino acid concentrations, and enzyme activity (19) also have been linked to *Epichloë* infection of drunken horse grass. While the colonization of drunken horse grass by *Epichloë* spp. and their beneficial traits is well documented, much less is known about the overall microbial composition of drunken horse grass (25).

In addition to these well-known effects, *Epichloë* spp. can also influence the underground activities of their host plants (26, 27). Comparative archaeal and bacterial gene expression in nitrogen metabolism showed significant differences in the nitrifying (*amoA*-AOB and *amoA*-AOA) and denitrifying (*nirS*, *nirK* and *nosZ*) genes in the rhizosphere soil of five drunken horse grass ecotypes (28). Large-scale microbiomic changes have also been observed, including an endophyte-induced increase of root-associated AM fungal diversity under drought conditions, consistent with AM fungal responses such as increased rhizosphere soil bacterial community, increased fungal diversity, and increased microbial biomass (26, 27, 29). However, a mechanism by which *Epichloë* spp. causes these responses in the host plant microhabitats of drunken horse grass remains elusive.

Epichloë symbiosis affects various host fitness-related traits, including agronomic properties, stand persistence, competitive ability, drought and low-nutrient tolerance, and resistance to herbivores via production of fungal alkaloids (30). Endophyte symbiosis causes changes in chemical and alkaloid content by altering some metabolic pathways within the host plant, and these changes may be transferred from fresh to senescent shoots (31), and thus may directly alter litter quality (32, 33). The presence of endophyte impacts the decomposer community, such as the components of detrital food webs, including nematodes, earthworms, mites, collembola and soil microflora (34, 35). Soil fungi and bacteria may also be influenced by litter containing endophyte and

may be modulated by root exudates of the host plant (36). However, we know very little at the microbiome and metabolic levels about the effect of endophyte symbiosis on host microhabitats. Among the unknowns are the dynamics of microorganisms from host shoot and root tissues during plant growth, the metabolic processes that are engaged during microbial changes, and regulatory relationships between root microbes and root exudates. Knowledge of the latter is important if we hope to clarify the effect of foliar endophytes on leaf, root and soil microorganisms in drunken horse grass via integrating microbiology and metabolomics analysis. The presence of *Epichloë* have been implicated in the mediation of host metabolism in the model plant *Lolium perenne* and other species (37-39). However, we are far from a comprehensive understanding of the biosynthesis, diversity, and relevance of specialized metabolites for biotic and abiotic responses. More broadly climatic adaptation in the *A. inebrians*-*Epichloë* symbiosis can provide knowledge use grass-endophyte symbiosis for conservation and agronomic purposes.

The goal of this work was to monitor changes in shoot and root microbes and root exudates utilizing a constant genetic background of drunken horse grass with and without *E. gansuensis* (I=endophyte containing; F=endophyte free) under ammonia-N (A) treatment in order to identify microbes and metabolites that are responsive to the *Epichloë* status of the host plant. The results revealed that *Epichloë* infection produced major shifts in bacterial community composition in shoots and roots of *A. inebrians* under treatments: water (CK) and ammonium nitrogen ($\text{NH}_4^+\text{-N}$). We used metabolite analyses to investigate specialized metabolism responses to ammonia-N (A) treatment in I and F plants. The results of this work are presented here.

RESULTS

Differential assemblage of shoot and root-associated bacterial microbiota in response to the presence or absence of *E. gansuensis*. We first investigated the influence of symbiotic *E. gansuensis* and N addition on richness and diversity of bacterial communities in shoots and roots of the host plant *A. inebrians*. Both observed composition and α -diversity of bacterial communities on phylum level to host shoots and roots was affected by the endophyte infection under ammonia-N (A) treatment (Fig.1 and Fig.2). The number of unique species on genus level of endophyte-infected plants was up to 11-fold in shoots and 1.8-fold in roots over that of endophyte-free plants (Fig.S1b and d). The relative abundance histogram showed that Cyanobacteria was the dominant phylum of shoot-inhabiting bacteria of *A. inebrians* plants under ammonia-N treatment relative to the add-water control (Fig.1a). In roots, Proteobacteria and Bacteroidota were the dominant phyla in

endophyte-infected plants, the Firmicutes was the most abundant phylum in endophyte-free plants (Fig.1b). To determine whether ammonia-N (A) nitrogen drives the divergence in bacterial community composition between endophyte-infection and endophyte-free plants, we use Wilcoxon rank-sum test (Fig.1c to e). The plants infected with *E. gansuensis* all had significantly greater relative abundance of Proteobacteria ($P=0.0304$), Bacteroidota ($P=0.0304$), Actinobacteriota ($P=0.0304$), Bdellovibrionota ($P=0.0265$), Patescibacteria ($P=0.0294$) and Acidobacteriota ($P=0.0211$) in the shoots than the non-infected plants under A-N treatment (Fig.1e) but showed no significant difference with water treatment (Fig.1c). Compared with the endophyte-infected plants, endophyte-free plants had the higher relative abundance of Acidobacteriota and Patescibacteria in roots under the control treatment (W) (Fig.1d). With the A-N treatment, the *E. gansuensis* infection significantly increased the relative abundance of Proteobacteria in root from 24.63% to 42.18% relative to endophyte-free plants, and significantly increased the relative abundance of Bacteroidota from 4.72% to 37.47%, Actinobacteriota from 1.27% to 5.63%, Bdellovibrionota from 0.06% to 1.12%, and also significantly increased the relative abundance of Fibrobacterota, Myxococcota and Armatimonadota, respectively (Fig.1f). However, the relative abundance of Firmicutes (65.50% to 2.94%) was significantly decreased in I roots than F roots under the A-N treatment (Fig.1f).

(Fig.1)

To further investigate the similarities or differences in bacterial community composition between endophyte-infected plants and endophyte-free plants, alpha diversity analysis and principal coordinate analysis (PCoA) were used. The *t*-test results showed that the α -diversity metrics (Shannon and Chao1 indexes) related to the shoot and root bacteria under A-N treatment was significantly increased by *Epichloe* infection but showed no significant differences with respect to water treatment (Fig.2a to d). Neither the bacterial community structure in shoots or roots was affected by the *Epichloe* symbiosis when water was supplied (Fig.2e and g). The bacterial community structure between I and F plants exhibited a significant difference in shoots (PERMANOVA: $R^2 = 0.4462$, $P = 0.027$) and in roots (PERMANOVA: $R^2 = 0.9103$, $P = 0.021$) under A-N treatment (Fig.2f to h).

(Fig.2)

Neither the relative abundance of abundant community OTUs nor the rare community OTUs in shoots and roots were

affected by the *Epichloe* symbiosis under water treatment (Fig.3a and c). The relative abundance of intermediate and rare OTUs in shoots were significantly increased in the plants that were infected with *E. gansuensis* under A-N treatment (Fig.3b). By contrast, however, the presence of *Epichloe* significantly reduced the rare OTUs relative abundance in roots under A-N treatment (Fig.3d).

(Fig.3)

***Achnatherum inebrians*-*E. gansuense* symbiont show N-inducible root metabolite alterations.** To

complement microbiological studies root metabolomes of endophyte-infected plants and endophyte-free plants under different forms of A-N and water treatments were examined using ultra-high performance liquid chromatography-tandem mass-spectrometry (UHPLC-MS/MS) analysis. We identified 3003 and 2991 metabolite features in positive and negative ion mode (QC) samples (Table S1). To compare metabolite profiles between endophyte-infected plants and endophyte-free plants under water and ammonia-N treatments after filtering of low-quality peaks, missing value padding, data normalization and RSD assessment of quality control 877 and 837 POS and NEG modes differential metabolites with names features were selected for downstream statistical analysis. With A-N treatment, 105 and 52 metabolic features were found to be differentially accumulated in roots of endophyte-infected plants and endophyte-free plants under POS and NEG modes, respectively. Not surprisingly, endophyte infection changed differential metabolite expression patterns under A-N treatment (Fig.4a and b), indicating that endophyte-infected plants had a quite different metabolite profile from that of endophyte-free plants. Hence, we further analyzed the metabolite profiles independently in roots between endophyte-infected plants and endophyte-free plants by a multivariate dimension-reduction orthogonal partial least squares discriminant analysis (OPLS-DA analysis) (Fig.4c and d). For both control conditions and A-N treatment, difference in metabolites between endophyte-infected plants and endophyte-free plants were separated, consistent with the result from volcano plot analysis (Fig.S2). Aligned with the microbiomic changes, Venn analysis of the untargeted metabolomic data showed that endophyte infection under ammonia-N treatment had a positive impact on metabolite composition (Fig.S3).

(Fig.4)

Our analysis of the top 30 metabolites contributing to the difference between I and F plants revealed that most compounds significantly increased in abundance by endophyte infection under A-N treatment (Fig.5b), whereas a significant decrease was

observed under water treatment compared to endophyte-free plants root (Fig.5a). Next, we set out to detect the signature metabolites where concentrations in roots of endophyte-infected plants were significantly higher than that in roots of endophyte-free plants. Among the annotated signature metabolites detected under water treatment, general classification metabolites were highly enriched in endophyte-infected plant roots (Fig.5a). In contrast, endophyte-infected plants root under ammonia-N treatment exhibited an enrichment of general metabolites for lipids and lipid-like molecules (Ethyl (4Z)-4,7-octadienoate), seven organic acids and derivatives (O-Phosphotyrosine, O-(2-fluoroethyl)-l-tyrosine, Tyrosyl-Leucine, (-)-Fumigaclavine B, Valyltryptophan, N1-(2-Methoxy-4-methylbenzyl)-n2-(2-(5-methylpyridin-2-yl) ethyl) oxalamide, and Isoleucyl-Valine), two organic oxygen compounds (Cis-Zeatin-7-N-glucoside and 2-Aminoacetophenone), organoheterocyclic compounds (Hydroxylated N-acetyl desmethyl frovatriptan), five unidentified compounds (Indirubin-3'-monoxime, (S)-3-([1,1'-Biphenyl]-4-yl)-2-aminopropanoic acid, 4-[(3-hydroxy-7-azaspiro[3.5]nonan-7-yl)methyl] benzonitrile, His Ser Val, and Chanoclavine-I) (Fig.5b). Furthermore, a few compounds identified by human metabolome database (HMDB) searches as alkaloids and derivatives (6-allyl-8b-Carboxy-ergoline, Cinchonidine and Norajmaline) and benzenoids ((2-Naphthalenyloxy) acetic acid and Zingerone) significantly accumulated in roots of endophyte-infected plants compared to endophyte-free plants (abundance all zero) under ammonia-N treatment (Supplementary Materials: MetaData and Fig.5b). Among the metabolites that were generally enriched in roots of endophyte-infected plants under ammonia-N supply includes ergometrine, which was increased by 732-fold over endophyte-free plant roots (Fig.5b). The results of a *t*-test showed that ergonovine content in shoot under water ($P=0.005$) and A-N($P<0.001$) treatments was significantly increased by endophyte infection (Fig.S4).

(Fig.5)

Endophyte symbiosis alters metabolic pathways in roots of *Achnatherum inebrians*. The pathway analysis revealed that the main metabolic pathways related to 164 compounds in roots of endophyte-infected plants and endophyte-free plants according to the KEGG database were purine metabolism ($P=0.004$) and vitamin B6 metabolism ($P=0.004$) under water treatment (Fig.6a). Compared to water treatment, the significantly enriched pathways for 157 metabolic compounds in E and F plants by the hypergeometric distribution algorithm and relative-betweenness centrality were phenylalanine metabolism ($P=0.002$), aminoacyl-tRNA biosynthesis ($P=0.030$), lysine degradation ($P=0.025$), tryptophan metabolism ($P=0.035$)

and cysteine and methionine metabolism ($P=0.036$) under A-N treatment (Fig.6b).

(Fig.6)

Visualization of biological and statistical correlation between bacterial community and root exudates using Sankey network. To provide evidence for endophyte effects on underground tissues and soil of host *A. inebrians*, we first assessed whether the root bacterial community was affected by root exudates in fungal endophyte-infected plants and N treatment. We wanted to evaluate whether endophyte symbiosis affects the interaction between bacteria and metabolites and if A-N treatment alters any correlations. Spearman correlation analysis was performed between root bacterial microbiota and metabolites of endophyte-infected plants and endophyte-free plants. According to the metabolites from microbiota, we identified three significant co-metabolism associated metabolic pathways that were associated with nutrient supply, that is, cysteine and methionine metabolism, tyrosine metabolism, arginine and proline metabolism. Take the cysteine and methionine metabolism, for example, five differential metabolites (i.e., S-Adenosylmethioninamine, spermidine synthase, adenosylmethionine decarboxylase, S-Adenosyl-L-methionine and 5'-Methylthioadenosine) were involved in the pathway, which participated in three different metabolic reactions (R00178, R01920 and R02869). Proteobacteria, Firmicutes, Actinobacteria and Bacteroidetes phyla were closely associated with metabolic reaction R00693 in the the Bio-Sankey network (Fig.7a). The relative abundance of S-Adenosylmethioninamine in root metabolism was significantly upregulated in endophyte-infected plants than endophyte-free plants under A-N treatment (Supplementary Materials: MetaData). The relative abundance of *Bacillus*, *Paenibacillus* and *Exiguobacterium* genera in roots were the most significantly increased in *A. inebrians* plants under A-N treatment (dark red if $P<0.05$), and were higher in endophyte-free plants compared to endophyte-infected plants, which were closely associated with metabolic reaction R01920 in parallel (Fig.7a). Statistical correlation analysis confirmed that Proteobacteria phylum were closely associated with S-Adenosylmethioninamine in the metabolic reaction R00693 by the STA-Sankey network (Fig.7b).

(Fig.7)

Co-Metabolism network shows differential metabolites shared by the host and microbiota, and their related microbes. The co-metabolism network of cysteine and methionine metabolism, tyrosine metabolism, arginine and proline metabolism showed that two root metabolites were associated with 49 differential root bacteria under A-N treatment

($P < 0.05$) (Fig.8). With the exception of very few bacteria, for example, *Enterococcus*, *Exiguobacterium*, *Paenibacillus* and *Fluviicola* genera, most of bacterial communities were significantly and positively correlated with S-Adenosylmethioninamine. We did, however, find a significantly positive effect of L-Dopa metabolites on *Bacills*, but there were the relationship between L-Dopa metabolites and he abundance of other bacterial community was mainly significant and negative (Fig.8). Bacteria-root metabolite association network provided more specific information on metabolic changes associated with *A. inebrians*-*E. gansuense* symbiont plants under A-N treatment.

(Fig.8)

DISCUSSION

Knowledge of the microbe-to-metabolite relationships in *Epichloe*-infected plants could help us understand specialized metabolic pathways that contribute to the shaping of the root microbiome. This knowledge could be used to improve the adaptability of host plants in deteriorating environments and reduce harvest losses (50). Extensive studies of *Epichloë* (e.g., *E. occultans*, *E. bromicola* and *E. festucae* var. *lolii*) have demonstrated that root and rhizosphere microbial communities in host plants show complex responses to endophyte symbiosis (51-53). By contrast, knowledge of the diversity of specialized metabolites in roots of perennial *A. inebrians* containing the *E. gansuense* symbiont and its relevance for root microbiome under ammonia-N (A) treatment is incomplete. Combined microbiome and metabolome analysis of plants bearing the fungal endophyte and non-infected control plants revealed common and distinct root metabolic alterations. Some specific metabolites identified in the root exudates of endophyte-infected plants, such as antioxidants (e.g., Glutaminylcysteine), growth regulator (e.g., (2-Naphthalenyloxy)acetic acid and Cis-Zeatin-7-N-glucoside) and alkaloids (plus alkaloid derivatives) (6-allyl-8b-Carboxy-ergoline, cinchonidine, ergometrine and norajmaline) may somehow relate to plant-microbe symbiosis, and could lead to increased absorption of bacteria from soil into plant tissues.

The current study demonstrated that changes in microbial communities in host plants might be driven by symbiosis with *Epichloe* (29), thus reflecting the role of fungal endophytes in the adaptation of *A. inebrians* plants. While prior studies showed that *E. gansuensis* -infected *A. inebrians* plants have a different microbiome composition and diversity within their phyllosphere, seed and root organs under optimal conditions (25, 39), the final bacterial community composition and diversity in

endophyte-infected plants and endophyte-free plants was at comparable levels under water treatment in our study, whereas bacterial abundance, community composition and diversity in shoots and roots were significantly different between endophyte-infected plants and endophyte-free plants under A-N treatment, resulting in a higher fold-change/increase of bacterial abundance and diversity index in endophyte-infected plants when compared to endophyte-free plants. These changes illustrated more pronounced N-induced endophyte effect, which was consistent with prior studies suggested the effect of fungal endophyte on host *A. inebrians* plants was susceptible to nitrogen (54). Rare microorganism plays important roles in regulating soil available nitrogen and plant biomass, and the uptake of nitrogen and carbon in environmental media (55). Nutrient-rich environments favoured by fast-growing bacteria (high-abundance bacteria), but nutrient scarcity choose from bacteria (low-abundance bacteria) with efficient nutrient utilization (56). Endophyte-infected plants have higher abundance of rare bacteria in shoots compared to shoots of endophyte-free plants in our study, whereas the abundance in roots of endophyte-infected plants was lower in endophyte-free plants under A-N treatment, as evidenced by the higher NO_3^- -N and NH_4^+ -N concentrations (Fig.S6).

Presence of known alkaloids in host plants with infected *Epichloë* endophyte supported substantial N responses in different grass-fungal endophyte symbioses during N treatment. Accumulation of ergometrine produced by *Achnatherum inebrians-E. gansuense* symbiont in shoot tissues was consistent with previously demonstrated endophyte-infected plants N responses (31, 57). The observed major contribution of pyrrolopyzine and ergot alkaloids to N absorption and conversion in shoots of endophyte-infected plants may be related to carbohydrates and amino acids accumulation as shown in, for example, endophyte-infected tall fescue (57), suggesting that A-N treatment induces endophyte effects on *A. inebrians* primary and secondary metabolism resulting in affects on microbial composition and diversity.

General and specialized metabolism pathways showed apparent metabolic differences between root of endophyte-infected plants and endophyte-free plants under A-N treatment. Accumulation of most organic acids, amino acids, and fatty acids in root exudates of *Achnatherum inebrians-E. gansuense* symbiont is consistent with previously demonstrated host plant low-N responses (54). Microbes could were attracted to root tips by root exudates (sugars, organic acids and amino acids), then was degraded by root-secreted reactive oxygen and also likely inducing electrolyte leakage in order to effectively extract nutrients by symbiotic microbes from microbes that colonize roots (58, 59). Our studies have demonstrated bioactive antioxidants (e.g.,

Glutamylcysteine) in roots exudates were significantly accumulated in endophyte-infected plants under A-N treatment, which protect against reactive oxygen species (ROS) that forms in the interactions between the endophytes and the plant cells, and would make the host better able to extract nutrients from the bacteria in roots. In addition, the observed major contribution of indirubin-3'-monoxime to metabolic alterations by endophyte infection may be related to inhibition of cell growth and multiplication of harmful organisms in soil solution of the rhizosphere (60). Additionally, a plant growth regulator with auxin like activity, (2-Naphthalenyloxy) acetic acid and Cis-Zeatin-7-N-glucoside were found in roots exudates of endophyte-infected plants that can stimulate the growth of plants and roots, as well as provide nutrients to intracellular and root microbes (61). Contrasting stress-related alkaloids in *A. inebrians* or other plants (54, 57) includes ergine and ergonovine that accumulate in response to abiotic stresses. Similarly, our metabolite analysis shows significant N-elicited ergometrine and other alkaloid species accumulation in root exudates of endophyte-infected *A. inebrians* plants.

In root exudates, except for the general classified metabolites, ergometrine, 6-allyl-8b-Carboxy-ergoline, cinchonidine and norajmaline alkaloids constituted the major determinants of metabolic differences between endophyte-infected plants and endophyte-free plants, especially under A-N treatment. All alkaloids are nitrogen-containing organic compounds having amino acids as the primary constituent in the biosynthetic pathway, such as lysergic acid have ergometrine as the basic material (62). Notably, A-N is converted by plants into amino acids and amides after absorbing, indicating that A-N has a greater effect on the biosynthesis of alkaloids. Ergometrine and ergine produced in the *A. inebrians*-*E. gansuense* symbiosis can improve plant resistance to predators (24), but also causes poisoning in livestock (5). Also, previous studies have demonstrated ergometrine in endophyte-infected plants of tall fescue that reduce the number of nematodes including *Helicotylenchus dihystera*, *Meloidogyne marglandi*, *Paratylenchus scribneri*, *Tylenchorhynchus acutus* and *P. projectus* around plant roots (63). The assembly of the root and soil microbiomes have been explored in some grass-endophyte symbiosis such as *A. inebrians*, *Lolium perenne* and *L. arundinaceum* (27, 64, 65). Prior studies that showed that root exudates are the main source of nutrients for rhizosphere microorganisms determining the ecological distribution and population composition of rhizosphere microorganism (66). Notably, our correlation analysis found a significant positive correlation between ergometrine content and most root bacterial microbial communities suggesting that endophyte infection may result in recruitment of bacterial communities (Fig.S7).

Significant N-induced accumulation of 6-allyl-8b-Carboxy-ergoline, cinchonidine and norajmaline alkaloids in roots of endophyte-infected plants supports that *Epichloë* symbiosis impacts root exudates. Although requiring further studies, higher abundance of these alkaloids produced only in root exudates of endophyte-infected plants under A-N treatment as compared to water treatment may result in increased nutrient absorption in *A. inebrians* plants. In addition, accumulation in roots of endophyte-infected plants support a role of these metabolites mediated by endophyte infection in the recruitment of microorganisms under A-N treatment. 6-allyl-8b-Carboxy-ergoline is a metabolite of ergoline, one of the three major families of ergot alkaloids. Ergot alkaloids from fungal endophyte-infected grasses could be responsible, at least in part, for the greater insect resistance of endophyte-infected grasses (67). Cinchonidine is a quinoline alkaloid with antibacterial effects (68). Norajmaline is ajmaline-sarpagine alkaloids and has effects on the central nervous system of animals (69). Interestingly, these alkaloids are only produced in roots of endophyte-infected plants under A-N treatment, and have different degrees of toxicity. There is a significant positive correlation between alkaloid content and relative abundance of root bacterial communities. However, the role of specialized metabolites such as alkaloids produced by endophyte symbiosis in these interactions between root microbiome and environmental factors is yet to be elucidated.

Plant endogenous microbes metabolite has homology and specificity with host plant, which can produce the same or similar secondary metabolites as the host, but also produce some compound which the host plant does not have (70). Similarly, this study showed significant N-induced accumulation of S-Adenosylmethioninamine only in roots of the *A. inebrians*-*E. gansuense* symbiosis supports a point about co-metabolism between endogenous bacteria and host plants (48). Further biological validation experiments are required to test whether specific bacteria are truly involved in metabolic reactions in plants containing the fungal endophyte, or a metabolite is utilized or produced by root bacteria, and to confirm their potential cause–consequence relationships.

In conclusion, these findings exemplify the advantages of combining microbiome and metabolome analysis to accelerate the discovery of specialized metabolites in root exudates of endophyte-infected plants and enable a deeper understanding of their relevance and role in recruiting beneficial bacterial communities. This approach revealed common and N-induced metabolic changes and the role of differential metabolites in the composition and diversity of root bacterial microbial communities in between *E. gansuensis*-infected and *E. gansuensis*-free *A. inebrians* plants. These insights provide the foundation for future

studies to investigate the diversity and function of ergot alkaloids and other specialized metabolites produced by *E. gansuensis* in *A. inebrians* plants.

MATERIALS AND METHODS

Plant material and treatments. Seeds of *Achnatherum inebrians* that were endophyte-free (F) and endophyte-infected with *Epichloë gansuensis* (I) were harvested from E and F fields, respectively, planted at Yuzhong Experimental Station of Lanzhou University, China, in 2017. All the seeds were stored at 4 °C in the dark until required. I and F seeds were planted in 72 holes plate of cultivation seedlings filled with vermiculite (2.0 kg), which was sterilized at 150 °C for 24 h. After 45 days, the presence of viable *Epichloë* in the subsequent seedlings was assessed by staining and microscopic examination. Endophyte-infected and endophyte-free seedlings were transferred into water (W) and an 0.1 mmol/L ammonia-N (A) nutrient solution. Plants were grown in a greenhouse at $25 \pm 2^\circ\text{C}$ with a 16 h: 8 h, light: dark photoperiod (c. $600 \mu\text{mol m}^{-2} \cdot \text{s}^{-1}$) in a randomised block design and at 8 wk post-cultivation. Root exudates were collected using a root exudate collection device after plants reached 7 days of growth. Leaf and root tissues of I and F plants (n = 4 per group) under different nutrient treatments were collected after 9 wk (8 wk+7 days) at a consistent time and immediately flash-frozen in liquid nitrogen.

Ergonovine determination. Ergonovine was extracted from 50mg freeze-dried grass material in two stages using a two-phase solvent system using the extraction method of Li et al. (20). Ergonovine was quantified on an Agilent 1100 series high performance liquid chromatography system (HPLC) (Agilent Technologies Inc., USA), with an 250 mm × 4.6 mm Eclipse XDB-C18 column (Agilent Technologies Inc., USA) containing 5 μm particles. Mobile phase: 0.1M NH₄OAc (A) : 0.1M CH₃CN=3:1, at a flow rate of 1.0 mL/min, 5μl injection volume, and open the wash needle (wash with 10% isodiol). The eluent gradients: 0.0min-10.0min:95% A:5% B; 10.0min-25.0min:75% A:25% B; 35.0min-50min:50% A:50% B; 51.0min-55.0min:5% A:95% B; 56.0min-71.0min: 95% A:5% B. The detection wavelength (Ex) of a fluorescence detector was set at 312 nm, the emission wave (Em) was 427nm, and the column temperature was 25°C. Ergonovine content was identified and quantified by comparison with external standard curves.

DNA extraction, PCR amplification and sequencing. Total DNA was extracted from 30 mg of leaves or roots of *Epichloë*-infected and *Epichloë*-free plants (n=6) either nitrogen-treated or well-watered (control) using a modified CTAB

DNA extraction procedure (40). The concentration and purity of extracted DNA was determined using a Micro Nanodrop ND-1000 UV-Vis Spectrophotometer (Nanodrop Technologies Inc., Wilmington, DE, USA), and DNA extract quality was checked on a 1% agarose gel. DNA amplification was achieved using a ABI GeneAmp[®] 9700 thermal cycler (Applied Biosystems[™] now part of Thermo Fisher Scientific Inc.) and TransStart Fastpfu DNA Polymerase (Beijing TransGen Biotech Co., Ltd. AP221-02) with primer sequences (338F: 5'-ACTCCTACGGGAGGCAGCAG-3' and 806R: 5'-GGACTACHVGGGTWTCTAAT-3') (41), and with fluorescence quantification using Quantus[™] Fluorometer (Promega Corp., USA). Four of the six biological replicates with the highest DNA quality were selected for *16S rRNA* gene sequencing. Preparation of PCR product libraries, PCR amplification and sequencing was performed by Majorbio Bio-Pharm Technology Co., Ltd. (Shanghai, China). In brief, following PCR amplification analysis and quantitation, purified PCR product libraries were generated using the NEBNext[®] Rapid DNA-Seq Kit (New England Biolabs Inc., USA) and sequenced on an Illumina MiSeq PE300 (Illumina Inc., San Diego, CA, USA) sequencing platform generating 20-60 million 413 bp paired-end reads per sample. The *16S rRNA* nucleotide sequence analysis was performed on the free online platform of Majorbio Cloud (www.majorbio.com) using the QIIME-1.9.1 pipeline (42) (<http://qiime.sourceforge.net/>). Briefly, low-quality reads (length<50 bp or with a quality value <20 or having N bases) were removed by fastp (43) (<https://github.com/OpenGene/fastp>, version 0.19.6). Filtered, high-quality PE reads were spliced according to overlap relationship using flash (<http://www.cbcb.umd.edu/software/flash>, version 1.2.11) and were clustered for operational taxonomic unit (OTU) based on 97% similarity using uparse (44) (<http://drive5.com/uparse/>, version 7.1), and chimeras were removed. Species taxonomic annotation with a confidence threshold of 70% was based on best matches to databases from Silva 16S rRNA gene (v138) using RDP classifier (<http://rdp.cme.msu.edu/>, version 2.11).

Metabolite extraction. Here, metabolites were extracted from a 100 μ L liquid sample with 400 μ L methanol: acetonitrile (1:1, v/v) solution by sonicating at 40 kHz for 30 min at 5°C. Samples were placed at -20°C for 30 min to precipitate proteins. The resultant samples were centrifuged for 15min (13000g, 4°C) to remove solid particles. The supernatant was carefully transferred to new microtubes and evaporated under a gentle stream of nitrogen. After desiccation, samples were reconstituted in 100 μ L loading solution of acetonitrile: water (1:1, v/v) by sonicating briefly in a 5°C water bath. Extracted metabolites were

centrifuged for 15 min (13000g, 4°C) and cleared supernatants was transferred into fresh vials and stored at -80°C prior to liquid chromatography-mass spectrometry (LC-MS) analysis.

UHPLC-MS/MS analysis. Ultra-high performance liquid chromatography-tandem mass-spectrometry (UHPLC-MS/MS) analysis was performed in positive and negative mode (3500V to -3500V) following the protocol of González et al. (45). In brief, metabolite analyses were performed using a High Mass Resolution quadrupole Orbitrap mass spectrometer (Thermo Q ExactiveTM HF-X, Thermo Scientific Inc, USA) equipped with electrospray ionization (ESI) source operating (Thermo Scientific Inc., , USA). Metabolite separation (2 µL sample) was conducted on a HSS T₃ column (100 mm × 2.1 mm, 1.8 µm; Waters) using 0.1% formic acid in water: acetonitrile (95:5, v/v) (solvent A) and 0.1% formic acid in acetonitrile : isopropanol : water (47.5:47.5:5, v/v) (solvent B) as mobile phase and the following parameters: flow rate of 0.4 ml min⁻¹; column temperature 40°C; 2 µL injection; method: 0-3.5 min (0% to 24.5% B; 0.4 ml min⁻¹), 3.5-5-5.5 min (24.5%, 65% to 100% B; 0.4 ml min⁻¹), 5.5-7.4 min (100% B; 0.4 to 0.6 ml min⁻¹), 7.4-7.6-7.8 min (100%, 51.5% to 0% B; 0.6, 0.6 to 0.5 ml min⁻¹) and 7.8-9-10 min (0% B, 0.5 to 0.4 and 0.4 ml min⁻¹). The optimal parameters: heater temperature of 425°C; capillary temperature of 325°C; sheath gas flow rate at 50 arb; aux gas flow rate at 13 arb; ion-spray voltage floating (ISVF). Mass spectra were acquired in continuum mode over m/z 70-1050 using data-dependent MS/MS acquisition (DDA), with normalized collision potential scanned between 20-40-60 V rolling for DDA. Full MS resolution was 60000, and MS/MS resolution was 7500. Quality control sample (QC) and reference samples were analyzed every 10 injections to monitor the stability of the LC-MS system.

Data processing and annotation. After the mass spectrometry detection was completed, the obtained DDA data were processed using PROGENESIS QI (v.3.0; Waters, Milford, MA, USA) using line baseline filterion, retention time (RT) alignment, mass correction, peak recognition and integration (filtering low quality peaks), adduct grouping and deconvolution and the following parameters: peak picking with the automatic sensitivity method: default settings; RT range: 0.1-8.0 min. Annotated metabolites were defined by RT and m/z information (designated as features). Human Metabolome (HMDB, <http://www.hmdb.ca/>), Kyoto Encyclopedia of Genes and Genomes (KEGG, <http://www.genome.jp/kegg/>), Metlin (<https://metlin.scripps.edu/>) and Majorbio databases were used to provide feature annotations based on 10 ppm precursor mass tolerance, 95% isotope similarity and 10 ppm fragment mass tolerance. Metabolic features detected at least 80 % in any set of

samples were retained. Before downstream statistical analyses, ion abundances of each of the metabolic features were normalized to the internal standard telmisartan by sum, based on four biological replicates. Meanwhile, variables with relative standard deviation (RSD) > 30% of QC samples were removed, and log₁₀ logarithmization was performed to obtain the final data matrix.

Differential metabolite mining. The normalized and transformed data were used for statistical analysis using the free online platform of majorbio cloud (46) (cloud.majorbio.com). Briefly, orthogonal least partial squares discriminant analysis (OPLS-DA) was conducted using ROPLS package (Version 1.6.2) in R, and afterward evaluated the stability of the model using 7-cycle interactive validation was evaluated. Differentially expressed metabolites were identified based on the variable importance in the projection (VIP) obtained by the OPLS-DA model, the *P*-value of student's *t*-test, fold change (FC) and the following parameters: VIP>1; *P*<0.05; FC<1 or FC>1, default. Significantly different metabolites between endophyte-infected and endophyte-free plants under different N treatment were summarized, and mapped into their biochemical pathways through metabolic enrichment and pathway analysis using the KEGG database. In addition, SCIPY.STATS (Python packages, <https://docs.scipy.org/doc/scipy/>) and Fisher's exact test (*P*<0.05) were used to identify statistically significantly enriched pathways. Different metabolites were classified based on the pathways in which they are produced or the functions they perform.

Statistical analyses. OTUs of all samples were classified as abundant (relative abundances > 0.1%), intermediate (0.01% < relative abundances < 0.1%) and rare (relative abundances < 0.01%) based on their relative abundances (47). All the statistical analyses were conducted in R version 3.3.1 (<http://www.r-project.org>) and heat maps of community composition were created using the VEGAN package. The differences in diversity index of alpha were tested by ANOVA followed by the independent sample *t*-test. Similarity of bacterial community structure in the underground and aboveground in *A. inebrians* plants was examined using principal coordinate analysis (PCoA) based on the bray-curtis distance algorithm. Differential microbial community structure was identified based on the results in analysis of similarities (ANOSIM) and permutational multivariate analysis of non-parametric variance tests. Significantly different abundances in bacterial taxa between endophyte-infected and endophyte-free plants were identified based on adjusted *P*-value (P_{adj}) < 0.05 and Wilcoxon rank-sum test as selection criteria at phylum level. Pearson's correlations between bacterial taxa and metabolites were calculated using the STATS package in R. The correlation between bacterial community structure and metabolite abundances were tested by partial Mantel tests using the

VEGAN package.

Sankey network analysis. We used Bio-Sankey network and STA-Sankey network analysis in MetOrigin software to mine the correlation between bacteria at different classification levels (i.e., at phylum, class, order, family and genus) and metabolites in a metabolic pathway (48). The left censored data quantile regression interpolation (QRILC) and percentage calculation methods were used to interpolate the missing values and normalize the data before the statistical analysis (49). The OTUs correlation matrix was calculated using Spearman's rank-based correlation coefficient (Spearman's r). Correlations were considered statistically robust when Spearman's $r > 0.65$ and $P < 0.05$. We analyzed potential bacteria that might participate in a metabolic reaction by the Bio-Sankey network. Metabolite, metabolic reaction, and associated bacteria at different taxonomic levels were connected by bands gray as background. Data of statistically significant correlations between bacteria and a metabolite would be analyzed and high-lighted in red and green color, indicating positive and negative correlations. Statistical correlation between bacteria and a metabolic reaction was explored using the STA-Sankey network. They would be highlighted in red or green color if there exists a biological relationship between bacteria and metabolic reaction by MetOrigin database searching. STA-Sankey was complementary to Bio-Sankey results and focuses on statistical correlations between bacteria and metabolites in real datasets, which may explain the potential relationships between them. The Bio-Sankey and STA-Sankey networks could visualize the biological and statistical correlations between bacteria and root exudates straightforwardly.

Compliance with ethics requirements. This study did not involve any studies with human or animal subjects.

Data availability. The Sequence data associated with this project have been deposited in the NCBI (Accession Number: SUB13084834).

Author contributions. YJJ, ZJC and CJL conceived the original research. YJJ and ZJC analyzed data analysis and wrote the manuscript. YJJ and ZJC conducted plant nitrogen incubation experiments and plant growing, plant harvesting, sampling, and sample processing. YJJ and ZJC performed microbiomic and metabolomic analysis. JFW, KM and CJL assisted in the writing of the original article draft and revision of manuscript.

ORCID.

Zhenjiang Chen <https://orcid.org/0000-0002-2833-5668>

James F. White <https://orcid.org/0000-0002-6780-7066>

Chunjie Li <https://orcid.org/0000-0002-3287-2140>

SUPPLEMENTAL MATERIAL

Supplemental material is available online only.

SUPPLEMENTAL FILE 1, PDF file, 1.9 MB.

SUPPLEMENTAL FILE 2, XLSX file, 0.7 MB.

ACKNOWLEDGMENTS

We thank the editor and anonymous reviewers for their valuable comments.

Financial support for this work was provided by the National Basic Research Program of China (2014CB138702), the National Science Foundation of China (32201445), the China Postdoctoral Science Foundation (2021M701525), Program for Changjiang Scholars and Innovative Research Team in University, China (IRT17R50), Gansu Provincial Youth Science and Technology Fund Program (22JR5RA532), Gansu Province Outstanding Doctoral Students Project (22JR5RA434) and 111 Project (B12002). The authors are thankful for support from USDA-NIFA Multistate Project W4147, and the New Jersey Agricultural Experiment Station.

We declare no conflicts of interest.

REFERENCES

1. Yao X, Chai Q, Chen TX, Chen ZJ, Wei XK, Bao GS, Song ML, Wei WR, Zhang XX, Li CJ, Nan ZB. 2019. Disturbance by grazing and the presence of rodents facilitates the dominance of the unpalatable grass *Achnatherum inebrians* in alpine meadows of northern China. *The Rangeland Journal* 41:301-312. <https://doi.org/10.1071/RJ18096>.
2. Chen L, Li XZ, Li CJ, Swoboda GA, Young CA, Sugawara K, Leuchtman A, Schardl CL. 2015. Two distinct *Epichloë* species symbiotic with *Achnatherum inebrians*, drunken horse grass. *Mycologia* 107:863-873. <https://doi.org/10.3852/15-019>.
3. Leuchtman A, Bacon CW, Schardl CL, White JF, Tadych M. 2014. Nomenclatural realignment of *Neotyphodium* species with genus *Epichloë*. *Mycologia* 106:202-215. <https://doi.org/10.3852/13-251>.
4. Li CJ, Nan ZB, Zhang CJ, Zhang CY, Zhang YH. 2009. Effects of drunken horse grass infected with endophyte on Chinese rabbit. *Journal of Agricultural Science and Technology* 11:84-90. (in Chinese).
5. Liang Y, Wang HC, Li CJ, Nan ZB, Li FD. 2017. Effects of feeding drunken horse grass infected with *Epichloë gansuensis* endophyte on animal performance, clinical symptoms and physiological parameters in sheep. *BMC Vet Re* 13:223-230. <https://doi.org/10.1186/s12917-017-1120-6>
6. Zhang XX, Li CJ, Nan ZB. 2011. Effects of cutting frequency and height on alkaloid production in endophyte-infected drunken horse grass (*Achnatherum inebrians*). *Sci China Life Sci* 54:567-571. <https://doi.org/10.1007/s11427-01-4181-y>.

7. Zhang XX, Li CJ, Nan ZB, Matthew C. 2012. *Neotyphodium* endophyte increases *Achnatherum inebrians* (drunken horse grass) resistance to herbivores and seed predators. *Weed Res* 52:70-78. <https://doi.org/10.1111/j.1365-3180.2011.00887.x>.
8. Zhang XX, Pei WY, Nan ZB. 2014. Antifungal activity of petroleum ether extracts from *Achnatherum inebrians* infected with *Neotyphodium gansuense*. *Sci China Life Sci* 57:1234-1235. <https://doi.org/10.1007/s11427-014-4660-z>.
9. Xia C, Li NN, Zhang XX, Feng Y, Christensen MJ, Nan ZB. 2016. An *Epichloë* endophyte improves photosynthetic ability and dry matter production of its host *Achnatherum inebrians* infected by *Blumeria graminis* under various soil water conditions. *Fungal Ecol* 22:26-34. <https://doi.org/1016/j.funeco.2016.04.002>.
10. Xia C, Christensen MJ, Zhang XX, Nan ZB. 2018. Effect of *Epichloë gansuensis* endophyte and transgenerational effects on the water use efficiency, nutrient and biomass accumulation of *Achnatherum inebrians* under soil water deficit. *Plant Soil* 424:555-571. <https://doi.org/10.1007/s11104-018-3561-5>.
11. Chen N, He RL, Chai Q, Li CJ, Nan ZB. 2016. Transcriptomic analyses giving insights into molecular regulation mechanisms involved in cold tolerance by *Epichloë* endophyte in seed germination of *Achnatherum inebrians*. *Plant Growth Regul* 80:367-375. <https://doi.org/10.1007/s10725-016-0177-8>.
12. Li CJ. 2005. Biological and ecological characteristics of *Achnatherum inebrians/Neotyphodium* endophyte symbiont. PhD dissertation, Lanzhou University, Lanzhou.
13. Zhong R, Zhang XX, Li CJ, Nan ZB. 2022. Vertically transmitted *Epichloë* systemic endophyte enhances drought tolerance of *Achnatherum inebrians* host plants through promoting photosynthesis and biomass accumulation. *J Fungi* 8:512-526. <https://doi.org/10.22541/au.164880184.46621958/v1>.
14. Wang JF, Nan ZB, Christensen MJ, Li CJ. 2018a. Glucose-6-phosphate dehydrogenase plays a vital role in *Achnatherum inebrians* plants host to *Epichloë gansuensis* by improving growth under nitrogen deficiency. *Plant Soil* 430:1-12. <https://doi.org/10.1007/s11104-018-3710-x>.
15. Wang JF, Nan ZB, Christensen MJ, Zhang XX, Tian P, Zhang ZX, Niu XL, Gao P, Chen T, Ma LX. 2018b. Effect of *Epichloë gansuensis* endophyte on the nitrogen metabolism, nitrogen use efficiency, and stoichiometry of *Achnatherum inebrians* under nitrogen limitation. *J Agric Food Chem* 66:4022-4031. <https://doi.org/10.1021/acs.jafc.7b06158>.
16. Liu YL, Hou WP, Jin J, Christensen MJ, Gu LJ, Cheng C, Wang JF. 2021. *Epichloë gansuensis* increases the tolerance of *Achnatherum inebrians* to low-P stress by modulating amino acids metabolism and phosphorus utilization efficiency. *J Fungi* 7:390-413. <https://doi.org/10.3390/JOF7050390>.
17. Yao X, Christensen MJ, Bao GS, Zhang CP, Li XZ, Li CJ, Nan ZB. 2015. A toxic endophyte-infected grass helps reverse degradation and loss of biodiversity of over-grazed grasslands in northwest China. *Sci Rep-UK* 5:18527-18535. <https://doi.org/10.1038/srep18527>.
18. Zhang XX, Li CJ, Nan ZB. 2010a. Effects of cadmium stress on seed germination, seedling growth and antioxidative enzymes in *Achnatherum inebrians* plants infected with a *Neotyphodium* endophyte. *Plant Growth Regul* 60:91-97. <https://doi.org/10.1007/s10725-009-9422-8>.
19. Zhang XX, Li CJ, Nan ZB. 2010b. Effects of cadmium stress on growth and anti-oxidative systems in *Achnatherum inebrians* symbiotic with *Neotyphodium gansuense*. *J Hazard Mater* 175:703-709. <https://doi.org/10.1016/j.jhazmat.2009.10.066>.
20. Li CJ, Nan ZB, Schardl CL. 2006. Levels and temporal variation of ergot alkaloids in endophyte-infected drunken horse grass, *Achnatherum inebrians*, in China. In: APS, CPS and MSA Joint Meeting Abstracts, Quebec City, Canada, pp.203-204.
21. Xia C, Zhang XX, Christensen MJ, Nan ZB, Li CJ. 2015. *Epichloë* endophyte affects the ability of powdery mildew (*Blumeria graminis*) to colonise drunken horse grass (*Achnatherum inebrians*). *Fungal Ecol* 16:26-33. <https://doi.org/10.1016/j.funeco.2015.02.003>.
22. Yao X, Fan YB, Chai Q, Johnson RD, Nan ZB, Li CJ. 2016. Modification of susceptible and toxic herbs on grassland disease. *Sci Rep-UK* 7:30635-3042. <https://doi.org/10.1038/srep30635>.
23. He YL, Chen TX, Zhang HJ, White JF, Li CJ. 2022. Fungal endophytes help grasses to tolerate sap-sucking herbivores through a hormone-signaling system. *J Plant Growth Regul* 41:2122-2137. <https://doi.org/10.22541/au.164864317.71161847/v1>.
24. Zhang XX, Li CJ, Nan ZB, Matthew C. 2012. *Neotyphodium* endophyte increases *Achnatherum inebrians* (drunken horse grass) resistance to herbivores and seed predators. *Weed Res* 52:70-78. <https://doi.org/10.1111/j.1365-3180.2011.00887.x>.
25. Liu BW, Ju YW, Xia C, Zhong R, Christensen MJ, Zhang XX, Nan ZB. 2022. The effect of *Epichloë* endophyte on

- phyllosphere microbes and leaf metabolites in *Achnatherum inebrians*. *iScience* 25:104144-104166. <https://doi.org/10.1016/j.isci.2022.104144>.
26. Yao X, Chen ZJ, Wei XK, Chen SH, White JF, Huang X, Li CJ, Nan ZB. 2020. A toxic grass *Achnatherum inebrians* serves as a diversity refuge for the soil fungal community in rangelands of northern China. *Plant Soil* 448:425-438. <https://doi.org/10.1007/s11104-020-04440-4>.
 27. Zhong R, Xia C, Ju YW, Zhang XX, Duan TY, Nan ZB, Li CJ. 2021. A foliar *Epichloë* endophyte and soil moisture modified belowground arbuscular mycorrhizal fungal biodiversity associated with *Achnatherum inebrians*. *Plant Soil* 458:105-122. <https://doi.org/10.1007/s11104-020-04537-w>.
 28. Jin YY, Chen ZJ, He YL, White JF, Malik K, Chen TX, Li CJ. 2022. Effects of *Achnatherum inebrians* ecotypes and endophyte status on plant growth, plant nutrient, soil fertility and soil microbial community. *Soil Sci Soc Am J* 84:1028-1042. <https://doi.org/10.1002/saj2.20420>.
 29. Ju YW, Zhong R, Christensen MJ, Zhang XX. 2020. Effects of *Epichloë gansuensis* endophyte on the root and rhizosphere soil bacteria of *Achnatherum inebrians* under different moisture conditions. *Front Microbiol* 11:747-760. <https://doi.org/10.3389/fmicb.2020.00747>.
 30. Gibert A, Volaire F, Barre P, Hazard L. 2012. A fungal endophyte reinforces population adaptive differentiation in its host grass species. *New Phytol* 194:561-571. <https://doi.org/10.1111/j.1469-8137.2012.04073.x>.
 31. Hunt, M. G., Rasmussen, S., Newton, P. C. D., Parsons, A. J., Newman, J. A., 2010. Near-term impacts of elevated CO₂, nitrogen and fungal endophyte-infection on *Lolium Perenne* L. growth, chemical composition and alkaloid production. *Plant Cell Environ* 28:1345-1354. <https://doi.org/10.1111/j.1365-3040.2005.01367.x>.
 32. Chen ZJ, Jin YY, Yao X, Wei XK, Li XZ, Li CJ, White JF, Nan ZB. 2021. Gene analysis reveals that leaf litter from *Epichloë* endophyte-infected perennial ryegrass alters diversity and abundance of soil microbes involved in nitrification and denitrification. *Soil Biol Biochem* 154:108123. <https://doi.org/10.1016/j.soilbio.2020.108123>.
 33. Omacini M, Chaneton EJ, Ghersa CM, Otero P. 2004. Do foliar endophytes affect grass litter decomposition? A microcosm approach using *Lolium multiflorum*. *Oikos* 104:581-590. <https://doi.org/10.1111/j.0030-1299.2004.12915.x>.
 34. Lemons A, Clay K, Rudgers JA. 2005. Connecting plant-microbial interactions above and belowground: a fungal endophyte affects decomposition. *Oecologia* 145:595-604. <https://doi.org/10.1007/s00442-005-0163-8>.
 35. Purahong W, Hyde KD. 2011. Effects of fungal endophytes on grass and non-grass litter decomposition rates. *Fungal Divers* 47:1-7. <https://doi.org/10.1007/s13225-010-0083-8>.
 36. Omacini M, Semmartin M, Pérez LI, Gundel PE. 2012. Grass-endophyte symbiosis: A neglected aboveground interaction with multiple belowground consequences. *Appl Soil Ecol* 61:273-279. <https://doi.org/10.1016/j.apsoil.2011.10.012>.
 37. Dupont PY, Eaton CJ, Wargent JJ, Fechtner S, Solomon P, Schmid J, Day RC, Scott B, Cox MP. 2015. Fungal endophyte infection of ryegrass reprograms host metabolism and alters development. *New Phytol* 208:1227-1240. <https://doi.org/10.1111/nph.13614>.
 38. Green KA, Berry D, Feussner K, Eaton CJ, Ram A, Mesarich, CH, Solomon P, Feussner I, Scott B. 2020. *Lolium perenne* apoplast metabolomics for identification of novel metabolites produced by the symbiotic fungus *Epichloë festucae*. *New Phytol* 227:559-571. <https://doi.org/10.1111/nph.16528>.
 39. Liang JJ, Gao GY, Zhong R, Liu BW, Christensen MJ, Ju YW, Zhang W, Wu Z, Li C, Li CJ, Nan ZB. 2023. Effect of *Epichloë gansuensis* endophyte on seed-borne microbes and seed metabolites in *Achnatherum inebrians*. *Microbiol Spectr* 11. <https://doi.org/10.1128/spectrum.01350-22>.
 40. Mundy DC, Vanga BR, Thompson S, Bulman S. 2018. Assessment of sampling and DNA extraction methods for identification of grapevine trunk microorganisms using metabarcoding. *New Zealand Plant Protection* 71:10-18. <https://doi.org/10.30843/nzpp.2018.71.159>.
 41. Liu CS, Zhao DF, Ma WJ, Guo YD, Wang AJ, Wang QL, Lee DJ. 2016. Denitrifying sulfide removal process on high-salinity wastewaters in the presence of *Halomonas* sp. *Appl Microbiol Biot* 100:1421-1426. <https://doi.org/10.1007/s00253-015-7039-6>.
 42. Caporaso JG, Kuczynski J, Stombaugh J, Bittinger K, Bushman FD, Costello EK, Fierer N, Peña AG, Goodrich JK, Gordon JI, Huttley GA, Kelley ST, Knights D, Koenig JE, Ley RE, Lozupone CA, McDonald D, Muegge BD, Pirrung M, Reeder J, Sevinsky JR, Turnbaugh PJ, Walters WA, Widmann J, Yatsunencko T, Zaneveld J, Knight R. 2010. QIIME allows analysis of high-throughput community sequencing data. *Nat Methods* 7:335-336. <https://doi.org/10.1038/nmeth.f.303>.

43. Chen SF, Zhou YQ, Chen YR, Gu J. 2018. fastp : an ultra-fast all-in-one FASTQ preprocessor. *Bioinformatics* 34:i884-i890. <https://doi.org/10.1093/bioinformatics/bty560>.
44. Edgar RC. 2013. UPARSE: highly accurate OTU sequences from microbial amplicon reads. *Nat Methods* 10:996-998. <https://doi.org/10.1038/nmeth.2604>.
45. González RR, Fernández RF, Vidal JLM, Frenich AG, Pérezmg. 2011. Development and validation of an ultra-high performance liquid chromatography–tandem mass-spectrometry (UHPLC–MS/MS) method for the simultaneous determination of neurotransmitters in rat brain samples. *J Neurosc Meth* 198:187-194. <https://doi.org/10.1016/j.jneumeth.2011.03.023>.
46. Ren Y, Yu G, Shi, CP, Liu LM, Guo, Q, Han C, Zhang D, Zhang L, Liu BX, Gao H, Zeng J, Zhou Y, Qiu YH, Wei J, Luo YC, Zhu FJ, Li XJ, Wu Q, Li B, Fu WY, Tong YL, Meng J, Fang YH, Dong J, Feng YT, Xie SC, Yang QQ, Yang H, Wang Y, Zhang JB, Gu HD, Xuan HD, Zou GQ, Luo C, Huang L, Yang B, Dong YC, Zhao JH, Han JC, Zhang XL, Huang HS. 2022. Majorbio Cloud: A one-stop, comprehensive bioinformatic platform for multi-omics analyses. *iMeta* 1:e12-e19. <https://doi.org/10.1002/imt2.12>.
47. Zheng W, Zhao ZY, Lv FL, Wang R, Wang ZH, Zhao ZY, Li ZY, Zhai BN. 2021. Assembly of abundant and rare bacterial and fungal sub-communities in different soil aggregate sizes in an apple orchard treated with cover crop and fertilizer. *Soil Biol Biochem* 156:108222. <https://doi.org/10.1016/j.soilbio.2021.108222>.
48. Yu G, Xu CF, Zhang DN, Ju F, Ni Y. 2022. MetOrigin: Discriminating the origins of microbial metabolites for integrative analysis of the gut microbiome and metabolome. *iMeta* 1:e10-e23. <https://doi.org/10.1002/imt2.10>.
49. Wei RM, Wang JY, Su MM, Jia E, Chen SQ, Chen TL, Yan N. 2018. Missing value imputation approach for mass spectrometry-based metabolomics data. *Sci Rep-UK* 8:663-673. <https://doi.org/10.1101/171967>.
50. Zhalnina K, Louie KB, Hao Z, Mansoori N, da Rocha UN, Shi SJ, Cho H, Karaoz U, Loqué D, Bowen BP, Firestone MK, Northen TR, Brodie EL. 2018. Dynamic root exudate chemistry and microbial substrate preferences drive patterns in rhizosphere microbial community assembly. *Nat Microbiol* 3:470-480. <https://doi.org/10.1038/s41564-018-0129-3>.
51. Chen J, Deng YK, Yu XH, Wu GH, Gao YB, Ren AZ. 2022a. *Epichloë* endophyte infection changes the root endosphere microbial community composition of *Leymus chinensis* under both potted and field growth conditions. *Microb Ecol* 85:1-13. <https://doi.org/10.1007/s00248-022-01983-0>.
52. Roberts E, Ferraro AR. 2015. Rhizosphere microbiome selection by *Epichloë* endophytes of *Festuca arundinacea*. *Plant Soil* 396:229-239. <https://doi.org/10.1007/s11104-015-2585-3>.
53. Wakelin S, Harrison S, Mander C, Dignam B, Rasmussen S, Monk S, Fraser K, O'Callaghan M. 2015. Impacts of endophyte infection of ryegrass on rhizosphere metabolome and microbial community. *Crop Pasture Sci* 66:1049-1057. <https://doi.org/10.1071/CP14321>.
54. Hou WP, Wang JF, Christensen J, Liu J, Zhang YQ, Liu YL, Cheng C. 2021. Metabolomics insights into the mechanism by which *Epichloë gansuensis* endophyte increased *Achnatherum inebrians* tolerance to low nitrogen stress. *Plant Soil* 463:1-22. <https://doi.org/10.1007/s11104-021-04930-z>.
55. Zhang Y, Dong SK, Gao QZ, Ganjurjav H, Wang XX, Geng W. 2019. “Rare biosphere” plays important roles in regulating soil available nitrogen and plant biomass in alpine grassland ecosystems under climate changes. *Agr Ecosyst Environ* 279:187-193. <https://doi.org/10.1016/j.agee.2018.11.025>.
56. Dai TJ, Wen DH, Bates CT, Wu LW, Guo X, Liu S, Su YF, Lei JS, Zhou JZ, Yang YF. 2022. Nutrient supply controls the linkage between species abundance and ecological interactions in marine bacterial communities. *Nat Commun* 13:175-184. <https://doi.org/10.1038/s41467-021-27857-6>.
57. Ryan GD, Shukla K, Rasmussen S, Shelp BJ, Newman SJA. 2014. Phloem phytochemistry and aphid responses to elevated CO₂, nitrogen fertilization and endophyte infection. *Agr Forest Entomon* 16:273-283. <https://doi.org/10.1111/afe.12055>.
58. Chang XQ, Kingsley KL, White JF. 2021. Chemical interactions at the interface of plant root hair cells and intracellular bacteria. *Microorganisms* 9:1041-1063. <https://doi.org/10.3390/microorganisms9051041>.
59. White JF, Kingsley KL, Verma SK, Kowalski KP. 2018. Rhizophagy cycle: an oxidative process in plants for nutrient extraction from symbiotic microbes. *Microorganisms* 6:95-115. <https://doi.org/10.3390/microorganisms6030095>.
60. Heiss EH, Schachner D, Donati M, Grojer CS, Dirsch VM. 2016. Increased aerobic glycolysis is important for the motility of activated VSMC and inhibited by indirubin-3 '-monoxime. *Vasc Pharmacol* 83:47-56. <https://doi.org/10.1016/j.vph.2016.05.002>.

61. Verma SK, Kingsley K, Irizarry I, Bergen M, Kharwar R.N, White JF. 2017. Seed-vectored endophytic bacteria modulate development of rice seedlings. *J Appl Microbiol* 122:1680-1691. <https://doi.org/10.13140/RG.2.2.17979.72482>.
62. Thamhesl M, Apfelthaler E, Schwartz-Zimmermann HE, Kunz-Vekiru E, Krska R, Kneifel W, Schatzmayr G, Moll EK. 2015. *Rhodococcus erythropolis* MTHt3 biotransforms ergopeptines to lysergic acid. *BMC Microbiol* 15:73-86. <https://doi.org/10.1186/s12866-015-0407-7>.
63. Latch GCM. 1993. Physiological interactions of endophytic fungi and their hosts. Biotic stress tolerance imparted to grasses by endophytes. *Agr Ecosyst Environ* 44:143-156. [https://doi.org/10.1016/0167-8809\(93\)90043-O](https://doi.org/10.1016/0167-8809(93)90043-O).
64. Chen ZJ, White JF, Malik K, Chen H, Jin YY, Yao X, Wei XK, Li XZ, Li CJ, Nan ZB. 2022b. Soil nutrient dynamics relate to *Epichloë* endophyte mutualism and nitrogen turnover in a low nitrogen environment. *Soil Biol Biochem* 174:108832. <https://doi.org/10.1016/j.soilbio.2022.108832>.
65. Guo J, McCulley RL, Phillips TD, Jr McNear DH. 2016. Fungal endophyte and tall fescue cultivar interact to differentially affect bulk and rhizosphere soil processes governing C and N cycling. *Soil Biol Biochem* 101:165-174. <https://doi.org/10.1016/j.soilbio.2016.07.014>.
66. Bais HP, Weir TL, Perry LG, Gilroy S, Vivanco JM. 2006. The role of root exudates in rhizosphere interactions with plants and other organisms. *Annu Rev Plant Biol* 57:233-266. <https://doi.org/10.1146/annurev.arplant.57.032905.105159>.
67. Clay K, Cheplick GP. 1989. Effect of ergot alkaloids from fungal endophyte-infected grasses on fall armyworm (*Spodoptera frugiperda*). *J Chem Ecol* 15:169-182. <https://doi.org/10.1007/BF02027781>.
68. Barennes H, Kahiatani F, Pussard E, Clavier F, Meynard D, Njifoutawou S, Verdier F. 1995. Intrarectal Quinimax (an association of Cinchona alkaloids) for the treatment of *Plasmodium falciparum* malaria in children in Niger: efficacy and pharmacokinetics. *T Roy Soc Trop Med H* 418:418-421. [https://doi.org/10.1016/0035-9203\(95\)90036-5](https://doi.org/10.1016/0035-9203(95)90036-5).
69. Taylor WI. 1965. Chapter 22 The ajmaline-sarpagine alkaloids. *the alkaloids: chemistry and physiology* 8:785-814. [https://doi.org/10.1016/S1876-0813\(08\)60061-5](https://doi.org/10.1016/S1876-0813(08)60061-5).
70. Pang ZQ, Chen J, Wang TH, Gao CS, Li ZM, Guo LT, Xu JP, Yi C. 2021. Linking plant secondary metabolites and plant microbiomes: a review. *Front Plant Sci* 12:621276-621798. <https://doi.org/10.3389/fpls.2021.621276>.

Figure Legends

Fig.1 (a and b) Bar plots showing the composition of the relatively abundant bacterial genera in shoot (a) and root (b) affected by endophyte infection under water (W) and ammonia-N (A) treatments. (c to f) Wilcoxon rank-sum test bar plots showing differential microbial communities analysis of the relatively abundant bacterial phyla in shoot (c and e) and root (d and f) between endophyte-infected plants (I) and endophyte-free plants (F) under water (W, c and d) and ammonia-N (A, e and f) treatments. X-axis: bacterial microbes at different phylum levels; Different colored boxes indicate endophyte-infected plants (I) and endophyte-free plants (F); Y-axis represents the average relative abundance of bacterial phyla between I and F plants.

Fig.2 (a to d) The independent sample t test for the effects of endophyte infection on alpha diversity of bacterial phyla in shoot (a and b) and root (c and d) under water (W) and ammonia-N (A) treatments. a, c: Shannon index; b, d: Chao 1 index. (e to h) Principal co-ordinates analysis (PCoA) showing similarity or difference in shoot (e and f) and root (g and h) of bacterial community composition between endophyte-infected plants (I) and endophyte-free plants (F) under water (W, e and g) and ammonia-N (A, f and h) treatments.

Fig.3 (a to d) Differential abundance analysis (DAA) of OUT relative abundance at abundant, intermediate and rare bacterial community in shoot (a and b) and root (c and d) between endophyte-infected plants (I) and endophyte-free plants (F) under water (W, a and c) and ammonia-N (A, b and d) treatments.

Fig.4 (a and b) Metabolic compounds that showed significant differently in root exudates of endophyte-infected plants (I) and endophyte-free plants (F) under water (W) and ammonia-N (A) treatments. a, b: Number of differentially accumulated metabolic features that were detected under POS (a) and NEG (b) modes, respectively, orange and green bars indicate the numbers of up-regulated and down-regulated metabolic features in relative abundance. (c and d) Orthogonal partial least-squares discriminant analysis (OPLS-DA) plots of ultra-high performance liquid chromatography-tandem mass-spectrometry (UHPLC-MS/MS) positive mode metabolome divergence between endophyte-infected plants (I) and endophyte-free plants (F) under water (W, c) and ammonia-N (A, d) treatments. x-axis: first predicted principal component explanatory degree; y-axis: first orthogonal component explanatory degree.

Fig.5 (a and b) VIP bar plot and clustered heat-map showing metabolite expression patterns and VIP of metabolites in multivariate statistical analysis in root exudates of endophyte-infected plants (I) and endophyte-free plants (F) under water (W, a) and ammonia-N (A, b) treatments. VIP_value: the contribution value of the metabolite to the difference between I and F plants. Left: metabolite VIP bubble plots. Y-axis: metabolites. X-axis: VIP value. The metabolites was arranged according to the size of the VIP value, from top to bottom. Right: metabolite expression heat-map. Heat-map columns: I and F plants. The red/blue color indicates the magnitude of the relative expression of the metabolites in root exudates of I and F plants, and the correspondence between the color gradient and the value is shown in the gradient color block. (c and h) Special differentially expressed metabolites between endophyte-infected plants (I) and endophyte-free plants (F). c: 6-allyl-8b-Carboxy-ergoline. d:

Cinchonidine. e: Ergometrine. f: Norajmaline. g: (2-Naphthalenyloxy)acetic acid. h: Cis-Zeatin-7-N-glucoside.

Fig.6 (a and b) KEGG topology analysis by relative-betweenness centrality methods showing pathways with significant metabolite enrichment in root exudates of endophyte-infected plants (I) and endophyte-free plants (F) under water (W, a) and ammonia-N (A, b) treatments. Each bubble: a KEGG Pathway. X-axis: the relative importance of metabolites in the pathway, impact value. Y-axis: the enrichment significance of the metabolite involvement pathway, $-\log_{10}(P \text{ value})$. Bubble size: impact value. The bubble with larger, the importance of pathway with greater.

Fig.7 (a) The BIO-Sankey Network for R01920 metabolic reaction in cysteine and methionine metabolism. (b) The STA-Sankey Network for R01920 metabolic reaction in cysteine and methionine metabolism.*: bacteria with statistically significant correlations with metabolites. Red/green nodes: up/down-regulation bacteria or metabolites. Red/green bands: the positive/negative correlations with metabolites. Dark red/green color: the statistically significance $P < 0.05$. The widths of the bands between nodes were linearly proportional to the number of bacteria; therefore, the wider band indicated the deeper involvement of a group of bacteria.

Fig.8 Co-Metabolism network shows differential metabolites shared by the host and microbiota, and their related microbes. Diamond and dot shapes: metabolites and microbes, correspondingly. Rectangular frames: metabolic pathways. Red/green nodes: up/down-regulation bacteria or metabolites. Red/green lines: the positive/negative correlations between microbes and metabolites.

Fig.1

Fig.2

Fig.3

Fig.4

Fig.5

Fig.6

Fig.7

a

BIO-ko00270: Cysteine and methionine metabolism (R01920)

b

STA-ko00270: Cysteine and methionine metabolism (R01920)

Fig.8

Supporting Information

Fig.S1 Venn plot showing number of common and unique OTUs in shoot (a and b) and root (c and d) of bacterial genera between endophyte-infected plants (I) and endophyte-free plants (F) under water (W, a and c) and ammonia-N (A, b and d) treatments.

Fig.S2 Volcano plot of metabolite expression differences between endophyte-infected plants (I) and endophyte-free plants (F) under water (W, a) and ammonia-N (A, b) treatments. The horizontal coordinate is the value of the fold change in the difference in expression of the metabolite between the two groups, i.e. \log_2FC . The vertical coordinate is the statistical test value for the difference in metabolite expression change, i.e. $-\log_{10}(p_value)$ value, with higher values indicating more significant expression differences. Each point in the graph represents a specific metabolite and the size of the point indicates the Vip value. The points on the left are metabolites with down-regulated expression differences and the points on the right are metabolites with up-regulated expression differences, with the more significant expression differences the further to the left, right and top the points are.

Fig.S3 Venn plot showing number of common and unique metabolite in root exudates of endophyte-infected plants (I) and endophyte-free plants (F) under water (W, a) and ammonia-N (A, b) treatments.

Fig.S4 Ergonovine content in shoot of endophyte-infected plants (I) and endophyte-free plants (F) under water (W) and ammonia-N (A) treatments.

Fig.S5 KEGG enrichment analysis showing a pathway of significant metabolite enrichment between endophyte-infected plants (I) and endophyte-free plants (F) under water (W) and ammonia-N (A) treatments. X-axis: the enrichment significance P -value. Y-axis: the KEGG pathway. The size of the bubbles represents how much of the pathway is enriched into the metabolic compound; the smaller the p -value, the more statistically significant it is, generally P -value less than 0.05 is considered as a significant enrichment term for this function.

Fig.S6 pH, and concentrations of NH_4^+ and NO_3^- in water (W) and ammonia-N (A) culture solutions for growth with endophyte-infected plants (I) and endophyte-free plants (F).

Fig.S7 Correlation heatmap plots visualizing the relationship between root bacterial microbial community and metabolic compounds of root exudates in endophyte-infected plants (I) and endophyte-free plants (F) under water (W, a) and

ammonia-N (A, b) treatments. X-axis: metabolites. Y-axis: bacterial phyla community. Different colors: correlation R-value.

If the *P*-value is less than 0.05, it is marked with *; right side legend, color intervals for different R values; *, $P \leq 0.05$; **, $P \leq 0.01$; ***, $P \leq 0.001$.

Table S1 Total ion count and identification statistics table.

Table S2 KEGG pathway between endophyte-infected plants (I) and endophyte-free plants (F) under water (W) and ammonia-N (A) treatments.

Table S3 HMDB compound classification between endophyte-infected plants (I) and endophyte-free plants (F) under water (W) and ammonia-N (A) treatments.

Excel table :Supplementary Materials: MetaData.

Fig.S1

Fig.S2

Fig.S3

Fig.S4

Fig.S5

Fig.S6

Fig.S7

Table S1 Total ion count and identification statistics table.

Ion mode	All peaks	Identified metabolites	Metabolites in Library	Metabolites in KEGG
pos	3003	580	519	159
neg	2991	198	157	71

Note: (1) Ion mode: the ion mode of the substance detected by the mass spectrometer, mainly: pos (positive ion mode) and neg (negative ion mode); (2) All peaks: the number of mass spectrometry peaks extracted by the software; (3) Identified metabolites: the number of metabolites finally identified by the primary and secondary mass spectrometry data (self-built libraries, Metlin, HMDB) (4) Metabolites in library: the number of metabolites annotated to public databases such as HMDB and Lipidmaps; (5) Metabolites in KEGG: the number of metabolites annotated to KEGG database.

Table S2 KEGG pathway.

First Category	Second Category	WI_WF	AI_AF
Metabolism	Amino acid metabolism	10	14
	Biosynthesis of other secondary metabolites	3	6
	Carbohydrate metabolism	1	0
	Lipid metabolism	2	6
	Metabolism of cofactors and vitamins	3	3
	Metabolism of other amino acids	1	2
	Metabolism of terpenoids and polyketides	2	2
	Nucleotide metabolism	4	3
Environmental Information Processing	Membrane transport	6	1
	Signal transduction	0	2
Genetic Information Processing	Translation	0	2

Table S3 HMDB compound classification

Superclass	WN		UN		AN		NN	
	Number	%	Number	%	Number	%	Number	%
Lipids and lipid-like molecules	40	27.78	32	18.39	35	26.72	18	38.3
Organic acids and derivatives	38	26.39	75	43.1	38	29.01	13	27.66
Organoheterocyclic compounds	23	15.97	26	14.94	20	15.27	8	17.02
Phenylpropanoids and polyketides	12	8.33	6	3.45	9	6.87	3	6.38
Benzenoids	11	7.64	13	7.47	8	6.11	1	2.13
Organic oxygen compounds	10	6.94	14	8.05	14	10.69	2	4.26
Nucleosides, nucleotides, and analogues	5	3.47	3	1.72	2	1.53	2	4.26
Alkaloids and derivatives	3	2.08	3	1.72	5	3.82		
Hydrocarbon derivatives	1	0.69	1	0.57				
Organic nitrogen compounds	1	0.69	1	0.57				

Revision Notes-Response to Editor Comments:

Dear Editor:

Thank you for your approval and constructive suggestions concerning our manuscript entitled "Interactions between *Epichloë* endophyte and the plant microbiome impact nitrogen responses in host *Achnatherum inebrians* plants (Spectrum02574-23)", which we submitted to Microbiology Spectrum. Those comments are all valuable in improving our manuscript. We have studied comments carefully and made revisions accordingly which we hope it meets with your approval.

Point-by-point responses to the issues raised by Editor:

Thank you for the privilege of reviewing your work. Below you will find my comments, instructions from the Spectrum editorial office.

Question: Despite the initial interest in the work, there are several issues with the presentation of the results, manuscript editing, and the selection of data to focus on in a single narrative. Although it may seem trivial, effectively communicating the results and conclusions is crucial; otherwise, poor narratives hinder the discovery of the manuscript's message. Therefore, to ensure a fair analysis of the manuscript, I suggest a thorough review of all sections. This includes rewriting the abstract and focusing on both the results and the discussion. The manuscript could benefit from considering only essential figures. In addition to these improvements, it is essential to present a well-formatted text and address the concerns raised by the reviewers.

Answer: According to your suggestions, we have rewritten the abstract, importance, introduction, discussion and conclusions, revised the materials and methods, presentation and results, reduced number of figures, and addressed the concerns raised by the reviewers. As following:

ABSTRACT

The clavicipitaceous fungus *Epichloë gansuensis* forms symbiotic associations with drunken horse grass (*Achnatherum inebrians*), providing biotic and abiotic stress protection to its host. However, it is unclear how *E. gansuensis* affects assembly of host plant-associated bacterial communities after ammonium nitrogen ($\text{NH}_4^+\text{-N}$) treatment. We examined the shoot- and root-associated bacterial microbiota, and root metabolites of *A. inebrians* when infected (I) or uninfected (F) with *E. gansuensis* endophyte. The results showed more pronounced $\text{NH}_4^+\text{-N}$ -induced microbial and metabolic changes in the endophyte-infected plants compared to the endophyte-free plants. *E. gansuensis* significantly altered bacterial community composition and β -diversity in shoots and roots, and increased bacterial α -diversity under $\text{NH}_4^+\text{-N}$ treatment. The relative abundance of 117 and 157 root metabolites significantly changed with *E. gansuensis* infection under water and $\text{NH}_4^+\text{-N}$ treatment compared to endophyte-free plants. Root bacterial community composition was significantly related to the abundance of the top 30 metabolites (VIP>2 and VIP>3) contributing to differences between I and F plants, especially alkaloids. The correlation network between root microbiome and metabolites was complex. Microorganisms in the Proteobacteria and Firmicutes phyla were significantly associated with the R00693 metabolic reaction of cysteine and methionine metabolism. Co-metabolism network analysis revealed common metabolites between host plants and microorganisms. (See in L 18-30).

IMPORTANCE

Our results suggest that the effect of endophyte infection is sensitive to nitrogen availability. Endophyte symbiosis altered the composition of shoot and root bacterial communities, increasing bacterial diversity. There was also a change in the class and relative abundance of metabolites. We found a complex co-occurrence network between root microorganisms and metabolites, with some metabolites shared between the host plant and its microbiome. The precise ecological function of the metabolites produced in response to

endophyte infection remains unknown. However, some of these compounds may facilitate plant-microbe symbiosis by increasing the uptake of beneficial soil bacteria into plant tissues. Overall, these findings advance our understanding of the interactions between the microbiome, metabolome, and endophyte symbiosis in grasses. The results provide critical insight into the mechanisms by which the plant microbiome responds to nutrient stress in the presence of fungal endophytes. (See in L 33-40).

INTRODUCTION

The perennial grass *Achnatherum inebrians* is highly adaptable, allowing it to dominate degraded grasslands across northwest China from 1,700-4,300 meters elevation. This 30-million-hectare range makes it an important species for controlling soil erosion and stabilizing sandy soils (1). However, its common symbiosis with the fungal endophytes *Epichloë gansuensis* and *E. inebrians* poses problems for livestock. These endophytes increase the plant's resilience, productivity and adaptability but produce toxic alkaloids (ergine and ergonovine) that can poison grazing animals (2, 3). Replacing infected grasses with endophyte-free cultivars is impractical because the fungi provide ecological benefits (4, 5). (See in L 45-50).

The advantages of the *Epichloë* symbiosis are well-studied. The fungi enhance the grass's drought and cold tolerance (6, 7), growth (8, 9), nutrient uptake (10-12), disease and pest resistance (13-16), tolerance of heavy metals (17, 18), and protecting biodiversity of over-grazed grasslands (19). They induce physiological and morphological changes in the host such as altered root-shoot ratios (8), increased photosynthesis (9), and shifts in sugars, amino acids, and enzymes (18). While the benefits of *Epichloë* endophyte to host health and growth are clear, the influence on above-ground and below-ground microorganisms is less understood. (See in L 51-56).

E. gansuensis infection altered the composition and diversity of bacterial and fungal community in phyllosphere and seed of *A. inebrians* in the field (20, 21), but there have been surprisingly few the effects of environmental components interacts with *Epichloë* endophyte on plant microbiome. Comparative studies reveal endophyte-related differences in root soil archaea and bacteria involved in nitrogen cycling (22). The fungi also increase root fungal diversity under drought and alter soil bacterial communities (23-25). However, the mechanisms behind these microbiome shifts are unknown. (See in L 57-61).

The symbiosis alters host metabolism by changing pathways and alkaloid production (26, 27). These changes affect litter decomposition and soil microflora (28-30). Foliar endophytes likely influence root microbes through both litter chemistry and root exudates (31). Nevertheless, we lack a comprehensive understanding of the specialized metabolites produced, their functions, and their microbial effects (32, 33). Among the unknowns are the dynamics of microorganisms from host shoot and root tissues during growth, the metabolic processes that are engaged during microbial changes, and regulatory relationships between root microbes and root exudates. Knowledge of the latter is important if we hope to clarify the effect of foliar endophytes on leaf, root and soil microorganisms in drunken horse grass via integrating microbiology and metabolomics analysis. (See in L62-68).

This knowledge gap motivates the current study. We will monitor changes in shoot and root microbes and exudates in *A. inebrians* with and without *E. gansuensis* under different nitrogen regimes. The goals are to identify responsive microorganisms and metabolites and elucidate how endophyte symbiosis affects plant-microbe interactions and specialized metabolism under varying nutrient availability. Overall, this introduction summarizes the current state of knowledge, identifies key gaps, and outlines an approach to advance our understanding of the complex interplay between the microbiome, metabolome, and symbiosis in this ecologically important grass. (See in L69-74).

MATERIALS AND METHODS

Plant material and treatment

Epichloë gansuensis endophyte-infected (I) and -free (F) *Achnatherum inebrians* seeds were collected from E+ and E- seed fields, and planted at Yuzhong Experimental Station of Lanzhou University, Gansu Province, China, in 2017. All the seeds were stored at 4°C in the dark until the following experiments. I and F seeds were planted in 72 holes plate of cultivation seedlings filled with vermiculite (2.0 kg), which was sterilized at 150°C for 24 h. After 45 days, endophyte status detection in seedlings was performed by

staining and microscopic examination following the method of Welty (34). All the I seedlings infected with endophyte and then 8 seedlings that differed only in the presence or absence of the *Epichloë* endophyte were selected for the following experiments, respectively. Endophyte-infected seedlings and endophyte-free seedlings were transferred into water (W) and 0.1 mmol/L ammonia-N (A) nutrient solutions. Plants were grown in a greenhouse at $25 \pm 2^\circ\text{C}$ with a 16 h: 8 h, light: dark photoperiod (c. $600 \mu\text{mol}\cdot\text{m}^{-2}\cdot\text{s}^{-1}$) in a random block design and at 8 wk post-cultivation. Root exudates were collected using a root exudates collection device after plants reached 7 days of growth. Leaf and root tissues of I and F plants ($n = 4$ per group) under different nutrient treatments were collected after 9 wk (8 wk+7 days) at a consistent time and immediately flash-frozen in liquid nitrogen (N_2). (See in L76-88).

Ergonvine extraction and determination

Alkaloid was extracted in two stages using a two-phase solvent system following a method modified from Miles et al. (35). In brief, the seedlings were frozen in the refrigerator at -20°C for 5 h and then dried in a freeze dryer at -60°C for 24 h. The freeze-dried material was ground to a fine powder with a mortar. A 50 mg powder sample was taken into 2 mL centrifuge tube, and 1 mL 20% ice acetic acid was added, followed by ultrasonication for 5 min, shaking for 2 min, and centrifugation for 5 min at 1000 r/min. The alkaloids extract was collected with 1 mL miscible liquids of 95% methanol and 5% ammonia after purification by filtration through a PCX column. The PCX columns were activated with 2 mL of methanol before use. The collected extract (0.25 mL) was filtered into a 1.5 mL brown chromatography vial through a $0.22 \mu\text{m}$ pore size membrane filter. The samples in the bottles were stored in the refrigerator at 4°C under dark conditions awaiting HPLC detection. (See in L90-97).

Ergonvine was quantified on an Agilent 1100 high performance liquid chromatography system (HPLC), with a $250 \text{ mm} \times 4.6 \text{ mm}$ Eclipse XDB-C18 column containing $5 \mu\text{m}$ particles. Mobile phase: 0.1M NH_4OAc (A): 0.1M CH_3CN =3:1, at a flow rate of 1.0 mL/min, $5 \mu\text{l}$ injection volume, and open the wash needle (wash with 10% isodiol). The eluent gradients: 0.0 min-10.0 min: 95% A: 5% B; 10.0 min-25.0 min: 75% A: 25% B; 35.0 min-50 min: 50% A: 50% B; 51.0 min-55.0 min: 5% A: 95% B; 56.0 min-71.0 min: 95% A: 5% B. The detection wavelength (Ex) of a fluorescence detector was set at 312 nm, the emission wave (Em) was 427 nm, and the column temperature was 25°C . Ergonovine was purchased from Sigma-Aldrich China. The 0 mg/L, 0.375 mg/L, 0.75 mg/L, 1.5 mg/L, 3 mg/L, 6 mg/L and 12 mg/L concentrations of ergometrine standard solutions were configured by diluting the mother liquor of the standard (20 mg/L). The relevant standard linear equations were established by the external standard method: $Y_{\text{Ergonovine}} = 154.98X - 10.152$, $R^2 = 0.9981$, Y: peak areas, X: Concentration (mg/L). The process was monitored and the peak area was determined using the Chem Station for LC Rev.A.10.01, USA. Alkaloid levels in injection volume was identified and quantified by comparing their peak areas with external standard curves. Ergonovine contents in plant samples was calculated according to the relevant equation and the dilution of the sample. (See in L98-110).

RESULTS

Differential assemblage of shoot and root-associated bacterial microbiota in response to the presence or absence of *Epichloë* endophyte

The findings of community composition analysis revealed that N addition had the greatest significant impact on the change of the plant microbiome (Fig S1 and Fig 1). Further, it was found that shoots microbiomes in shoots of endophyte-infected plants consisted mostly of Cyanobacteria, Proteobacteria, Bacteroidota and Actinobacteriota under nitrogen treatment, accounting for >90% of total relative abundance, with Cyanobacteria dominating the shoots-associated bacteria of endophyte-free plants (Fig 1 a). The root-associated bacterial communities of endophyte-infected plants, at the phylum level, were mostly dominated by Proteobacteria (41.71%), Bacteroidota (37.62%), Cyanobacteria (9.16%) and Actinobacteriota (5.64%), but for endophyte-free plants, Firmicutes (65.34%) and Proteobacteria (24.71%) was the most dominant phylum (Fig 1 b). It was observed that the *E. gansuensis* infection significantly increased the relative abundance of Proteobacteria in roots from 24.63% to 42.18% relative to endophyte-free plants, Bacteroidota from 4.72% to 37.47%, and Actinobacteriota from 1.27% to 5.63% (Fig 1 c to f). The rrn copy

number of rare communities were significantly changed by the presence of the endophyte under A-N treatment (Fig S2). (See in L204-215).

(Fig 1)

The Shannon index and Chao 1 index of bacterial communities in shoots and roots were significantly ($P < 0.05$) influenced by the presence of the *Epichloë* endophyte under N treatment, but showed no significant difference under water treatment (Fig 2 a to d). Meanwhile, principal co-ordinates analysis (PCoA) was used to visualize the β -diversity based on the Bray-Curtis dissimilarity, and ANOSIM analyses were conducted to evaluate the significance of changes in community structure ($P < 0.05$) between endophyte-infected plants and endophyte-free plants (Fig 2 e to h). (See in L217-221).

(Fig 2)

***Achnatherum inebrians*-*Epichloë gansuense* symbiont show N-inducible root metabolite alterations**

We identified 3003 and 2991 mass spectrometry peaks in positive and negative ion mode (QC) samples (Table S1). In total, 604 and 207 metabolites were successfully detected from I and F roots of *A. inebrians* in POS and NGE modes before data preprocessing, respectively. To eliminate or reduce the error caused by the analysis process, 580 and 198 POS and NEG modes differential metabolites with names features were selected for downstream statistical analysis after filtering of low-quality peaks, missing value padding, data normalization and RSD assessment of quality control (Table S1). Nitrogen treatment altered the ratio of the down-regulated and the up-regulated metabolites in root exudates in POS and NGE modes between endophyte-infected plants and endophyte-free plants (Fig 3 a, b and Fig S3). The OPLS-DA models results showed that these detected metabolites were significantly diverse between I and F root under A-N ($R^2X = 0.564$, $R^2Y = 0.992$, $Q^2Y = 0.855$) and water ($R^2X = 0.858$, $R^2Y = 0.999$, $Q^2Y = 0.924$) treatments (Fig 3 c and d). (See in L223-232).

(Fig 3)

Aligned with the microbiomic changes, the analysis revealed that *E. gansuense*-infected *A. inebrians* had a greater number of specific metabolites than *E. gansuense*-free *A. inebrians* under N treatment (Fig 4 b). These differential metabolites could be divided into 9 classes, including organic acids and derivatives, lipids and lipid-like molecules, organo-heterocyclic compounds, organic oxygen compounds, alkaloids and derivatives, and Benzenoids (SUPPLEMENTAL FILE 2 and Table S2). Our analysis of the top 30 metabolites ($VIP > 2$ and $VIP > 3$) contributing to the difference between I and F plants revealed that most compounds significantly increased in abundance by endophyte infection under A-N treatment (Fig 4 c), whereas a significant decrease was observed under water treatment compared to endophyte-free plants root (Fig 4 d). Among the up-annotated signature metabolites detected under water treatment, general classification metabolites were highly enriched (Fig 4 c). In contrast, except for the significant enrichment of general metabolites, some specific metabolites were also significantly enriched in the root of endophyte-infected plants root under A-N treatment (Fig 4 d and SUPPLEMENTAL FILE 2). The specific metabolites identified by human metabolome database (HMDB) searches as 6-allyl-8b-Carboxy-ergoline, Cinchonidine, Norajmaline and (2-Naphthalenyloxy) acetic acid were significantly accumulated in roots of endophyte-infected plants compared to endophyte-free plants (abundance all zero) under A-N treatment (Fig 4 e, f, h and i). In addition to ergometrine were significantly increased by 732-fold in roots of endophyte-infected plants than roots of endophyte-free plants (Fig 4g), the results of *t*-test showed that ergonovine content in shoot under water ($P = 0.005$) and A-N ($P < 0.001$) treatments was significantly increased by endophyte infection (Fig.S4). (See in L234-249).

(Fig.4)

There were 6 metabolite pathways significantly enriched ($P < 0.05$) in *A. inebrians* under water treatment (Fig S5 a). Compared to water treatment, the significantly enriched pathways for 157 metabolic compounds in I and F plants by the hypergeometric distribution algorithm and relative-betweenness centrality were phenylalanine metabolism ($P = 0.002$), aminoacyl-tRNA biosynthesis ($P = 0.030$), lysine

degradation ($P=0.025$), tryptophan metabolism ($P=0.035$) and cysteine and methionine metabolism ($P=0.036$) under A-N treatment (Fig S5 b). (See in L251-255).

Visualization of biological and statistical correlation between bacterial community and root exudates using Sankey network

According to the metabolites from microbiota, three significant co-metabolism associated metabolic pathways were identified associated with nutrient supply, including cysteine and methionine metabolism, tyrosine metabolism, and arginine and proline metabolism. For instance, five differential metabolites (i.e., S-Adenosylmethioninamine, spermidine synthase, adenosylmethionine decarboxylase, S-Adenosyl-L-methionine and 5`-Methylthioadenosine) were involved in the cysteine and methionine metabolic pathway, which participated in three different metabolic reactions (R00178, R01920 and R02869). Proteobacteria, Firmicutes, Actinobacteria and Bacteroidetes phyla were closely associated with metabolic reaction R00693 in the Bio-Sankey network (Fig.5 a). The relative abundance of S-Adenosylmethioninamine in root metabolism was significantly up-regulated in endophyte-infected plants than endophyte-free plants under A-N treatment (SUPPLEMENTAL FILE 2). The relative abundance of *Bacillus*, *Paenibacillus* and *Exiguobacterium* genera in roots was the most significantly increased in *A. inebrians* plants under A-N treatment (dark red if $P<0.05$), and was higher in endophyte-free plants compared to endophyte-infected plants, which were closely associated with metabolic reaction R01920 in parallel (Fig 5 a). Statistical correlation analysis confirmed that Proteobacteria phylum were closely associated with S-Adenosylmethioninamine in the metabolic reaction R00693 by the STA-Sankey network (Fig 5 b). (See in L256-270).

(Fig5)

Co-Metabolism network shows differential metabolites shared by the host and microbiota, and their related microbes

The co-metabolism network of cysteine and methionine metabolism, tyrosine metabolism, and arginine and proline metabolism showed that two root metabolites were associated with 49 differential root bacteria under A-N treatment ($P < 0.05$) (Fig 6). With the exception of very few bacteria, for example, *Enterococcus*, *Exiguobacterium*, *Paenibacillus* and *Fluviicola* genera, most of bacterial communities were significantly and positively correlated with S-Adenosylmethioninamine. However, we were able to find a significant positive effect of L-Dopa metabolites only on bacteria of the genus *Bacills*, but significant negative correlation with other bacterial community (Fig 6). Bacteria-root metabolite association network provided more specific information on metabolic changes associated with *A. inebrians*-*E. gansuense* symbiont plants under A-N treatment. (See in L272-281).

(Fig 6)

DISCUSSION

By performing biological and statistical correlation analyses between root bacteria and metabolites, this study provides novel evidence for the numerous close and intricate interrelationships between the two factors. The metabolomic data showed the *E. gansuensis* endophyte altered metabolic components in host plants. Integration of microbiome and metabolome data sheds light on the complex bidirectional interactions between endophytes, host plants, and microbial communities. Our multi-omic characterization of endophyte-infected *A. inebrians* advances understanding of the intricate network of signals mediating tripartite symbiosis. Further elucidation of the specific mechanisms underlying microbiome-metabolome interplay will be key. Overall, this systems biology approach coupling amplicon sequencing and metabolomics enables greater insight into the multidimensional impacts of endophytes on host-microbe interactions. (See in L283-290).

Effect of the *E. gansuensis* endophyte on the composition of plant bacterial microbial communities

The current study found that *A. inebrians* plants infected with the *E. gansuensis* endophyte harbored more complex phyllosphere microbial communities compared to uninfected plants (Fig 1a and 1b). These results align with previous studies by Liu et al. and Liang et al. showing altered microbiome composition and diversity in the phyllosphere, seeds, and roots of endophyte-infected *A. inebrians* under optimal conditions

(20, 21). Extensive research on other *Epichloë* endophytes like *E. occultans*, *E. bromicola*, and *E. festucae* var. *lolii* has also shown that root and rhizosphere microbiomes exhibit complex responses to endophyte symbiosis (46-48). Our findings further demonstrate that shifts in plant microbial communities can be driven by the presence of fungal endophytes (22, 25), reflecting the critical role endophytes play in host adaptation. (See in L292-298).

The current study demonstrated that the A-N treatment induced significant alterations in bacterial community composition in both shoots and roots of endophyte-infected plants compared to endophyte-free plants (Fig 1a and 1b). This manifested as a greater fold-change increase in bacterial abundance in infected versus uninfected plants under A-N conditions (Fig 1e and 1f). These findings illustrate an amplified nitrogen-responsive effect of the endophyte, aligning with previous research showing the influence of *Epichloë* endophytes on host plants is enhanced in nitrogen-rich environments (49). As proposed by prior studies, copiotrophic bacteria thrive in nutrient-rich conditions while oligotrophic bacteria with efficient nutrient utilization prevail in nutrient-poor settings (50). Accordingly, we found endophyte-infected shoots harbored more rare bacteria compared to endophyte-free shoots under A-N, while the opposite trend occurred in roots, mirroring measured nitrate and ammonium levels (Fig. S6). As rare microbes play key roles governing nitrogen availability and plant nutrient uptake (51), the endophyte-mediated shifts in low abundance bacteria may have significant ecological impacts. (See in L299-308).

Numerous studies demonstrate shoot and root microbiome alpha diversity critically regulates plant health, growth, and environmental adaptability (52, 53). Our work revealed the *Epichloë* endophyte markedly increased alpha and beta diversity of shoot and root bacteria under A-N but not under water-only conditions (Fig 2), aligning with previous findings (20, 21, 54). Taken together, these results suggest nitrogen availability enables endophytes to substantially augment the diversity and complexity of host-associated bacterial communities. Further investigation of the multifaceted interplay between nitrogen, endophytes, and microbiota will provide greater insight into the factors shaping holobiont resilience. Our integrative analysis of amplicon data with plant metabolomic and nutrient profiles represents a valuable systems biology approach to elucidating the intricate tripartite network of endophyte-host-microbe interactions. (See in L309-316).

Effect of the *E. gansuensis* endophyte on root metabolites

This study demonstrated that *E. gansuensis* infection was the primary driver of metabolic profile alterations in *A. inebrians*, as endophyte-infected and endophyte-free plants showed distinct profiles even under optimal conditions (21). Notably, we observed amplified nitrogen-responsive metabolic divergence between infected and uninfected plants (Fig 3), aligning with previous findings on low-nitrogen responses (49). General and specialized metabolism pathways exhibited pronounced differences between endophyte-infected and endophyte-free roots under A-N (Fig 4), mirroring the accumulation of various organic acids, amino acids, and fatty acids reported by Hou et al. (49). Other metabolomic studies reveal *Epichloë* endophytes can modulate host stress resilience, growth, and soil insect distributions by altering root exudate composition and metabolic pathways (55, 56). (See in L318-324).

We found *A. inebrians* alkaloid contents increased under salt and drought stresses, particularly cytotoxic ergonovine (57). Similarly, under A-N conditions, endophyte-infected plants showed significant ergometrine accumulation in root exudates (Fig S4) -contrasting related stress-induced alkaloids like ergine and ergonovine (49, 58). The major contribution of toxic pyrrolopyrazine and ergot alkaloids to nitrogen uptake and translocation in infected plant shoots may relate to the concurrent accumulation of carbohydrates and amino acids, as seen in endophyte-infected tall fescue (59). (See in L325-329).

Overall, our dual metabolomic and amplicon sequencing analyses provide novel insight into the close interrelationship between the endophyte-conferred alkaloid profile and the structure of the root microbiome. The significant correlations found between specific bacterial taxa and ergot alkaloids hint at a complex codependency. Further elucidation of the factors regulating alkaloid biosynthesis and secretion, the bioactivities of these compounds against different microbes, and subsequent microbial community shifts will

be key. This holistic multi-omic approach enables systems-level understanding of the multifaceted chemical signals mediating dynamic endophyte-host-microbe interactions belowground. (See in L330335).

Root microbial and metabolic interactions with *Achnatherum inebrians*-*Epichloë gansuense* symbiont

Prior studies show root exudates are the primary nutrient source for rhizosphere microbes, directly shaping ecological and compositional dynamics (60). Notably, correlation analysis revealed significant positive associations between root metabolite levels and bacterial taxa, suggesting endophyte infection may recruit communities by modulating metabolites (Fig.S7). Microbes chemotactically locate root tips using exuded sugars, organic acids, and amino acids but are then degraded by reactive oxygen species as the plant extracts nutrients from colonizers (61, 62). We found the antioxidant glutamylcysteine was markedly accumulated in endophyte-infected roots under A-N, potentially mitigating reactive oxygen and improving nutrient extraction from bacteria. (See in L336-343).

The co-occurrence network demonstrated significant interrelationships between the root microbiome and metabolome in *A. inebrians* (Fig 5). For instance, we observed nitrogen-induced accumulation of S-adenosylmethioninamine exclusively in endophyte-infected roots, highlighting metabolite co-metabolism between bacteria and the symbiotic holobiont (44). (See in L344-346).

Overall, integration of amplicon and metabolomic data provides greater insight into the chemical dialog regulating tripartite interactions belowground. Our findings reveal endophytes modify root exudate profiles to selectively enhance colonization by beneficial microbiota, highlighting the key role of plant-microbe signaling in symbiotic assembly and function. Further characterization of metabolite bioactivities, specialized microbial nutritive requirements, chemoattraction dynamics, and community interdependencies will elucidate the multifaceted mechanisms enabling robust, mutually beneficial endophyte-host-bacteria relationships. (See in L347-352).

Conclusions

Nitrogen significantly amplified the effects of the *Epichloë* endophyte on host-associated microbes and metabolites compared to water-only conditions. Our results demonstrate the *E. gansuense* endophyte induced substantial alterations in shoot and root bacterial community composition and beta diversity under NH_4^+ -N treatment. Relative abundance of rare taxa in shoots and alpha diversity in both shoots and roots were higher in infected versus uninfected plants under nitrogen fertilization. Concurrently, endophyte infection modulated root metabolite classes and abundances, with variations in alkaloid levels linked to shifts in the root microbiome assembly. Complex and significant correlations were found between root bacteria and metabolites, highlighting their multifaceted interplay. This integrated amplicon sequencing and metabolomics approach provides novel insight into the specialized metabolites secreted by endophyte-infected plants and their relevance in recruiting beneficial microbiota. Our findings exemplify the power of combinatorial multi-omics to elucidate the intricate chemical dialog mediating tripartite endophyte-host-microbe interactions belowground. Further characterization of metabolite bioactivities, chemoattraction mechanisms, and microbial dependencies will be key to deciphering the complex signaling enabling robust, mutually beneficial symbioses. Overall, this systems biology strategy represents a valuable tool for rapidly advancing discovery of bioactive compounds and clarifying their ecological roles. (See in L355-367).

Revision Notes-Response to Reviewer 1 Comments:

Dear the reviewers 1:

Thank you for your approval and constructive suggestions concerning our manuscript entitled "Interactions between *Epichloë* endophyte and the plant microbiome impact nitrogen responses in host *Achnatherum inebrians* plants (Spectrum02574-23)", which we submitted to Microbiology Spectrum. Those comments are all valuable in improving our manuscript. We have studied comments carefully and made revisions accordingly which we hope it meets with your approval.

Point-by-point responses to the issues raised by the reviewer 1:

Question # 1: Im sure there are some interesting results in this work but they are difficult to find under the serious problems concerning the English and grammar.

Answer: Thanks for your comments. We have changed the English and grammar on the abstract, importance, introduction, materials and methods, results, discussion and conclusions. One of our co-authors, a native English speaker, Professor James White (an expert working on interaction of plants and microbes) from Rutgers University of USA, and Dr. Kamran Malik from Lanzhou University of China has polished again the revisions. (See in L44-367 of Manuscript-JINyy-cleaned and updated version).

Question # 2: There is no explanation to how *Epichloë* was assessed in seedlings or how many seedlings contained the endophyte after assessment. This is a fundamental part of the experiments.

Answer: Asexual *Epichloë* symbionts are only vertically transmitted via the mother plant lineage, so the harvested endophyte-infected seeds are 100% carrier of *Epichloë* endophyte. During the experiments, plants were microscopically examined several times to confirm the presence or absence of the asexual *Epichloë* endophyte, we found that the all E+ seedlings infected with *Epichloë* endophyte.

According to your suggestions, we have changed "After 45 days, presence of the endophyte in seedling was tested by staining and microscopic examination." into "After 45 days, endophyte status detection in seedlings was performed by staining and microscopic examination following the method of Welty (34). All the I seedlings infected with endophyte and then 8 seedlings that differed only in the presence or absence of the *Epichloë* endophyte were selected for the following experiments, respectively." (See in L 80-83).

Question # 3: For the ergonovine determination, a reference (Li et al) is cited for the extraction method but this reference is a poster abstract and contains no valid information whatsoever.

Answer: We again checked the reference for the extraction method about ergonovine determination and indeed there is no specific information. However, the ergonovine alkaloid extraction method with we used was indeed modified by Chunjie Li based on the method of Miles et al. (1996).

According to your suggestions, we have changed "Li CJ, Nan ZB, Schardl CL. 2006. Levels and temporal variation of ergot alkaloids in endophyte-infected drunken horse grass, *Achnatherum inebrians*, in China. In:APS, CPS and MSA Joint Meeting Abstracts, Quebec City, Canada, pp.203-204." into "Miles CO, Lane GA, Menna ME. 1996. High levels of ergonovine and lysergic acid amide in toxic *Achnatherum inebrians* accompany infection by an *Acremonium* like endophytic fungus. J Agr Food Chem 5:1285-1290." (See in L 485-487).

We have changed "Alkaloid was extracted from 50mg freeze-dried material in two stages using a two-phase solvent system by extraction method of Li et al. (20)." into "Alkaloid was extracted in two stages using a two-phase solvent system following a method modified from Miles et al. (35). In brief, the seedlings were frozen in the refrigerator at -20°C for 5 h and then dried in a freeze dryer at -60°C for 24 h. The freeze-dried

material was ground to a fine powder with a mortar. A 50 mg powder sample was taken into 2 mL centrifuge tube, and 1 mL 20% ice acetic acid was added, followed by ultrasonication for 5 min, shaking for 2 min, and centrifugation for 5 min at 1000 r/min. The alkaloids extract was collected with 1 mL miscible liquids of 95% methanol and 5% ammonia after purification by filtration through a PCX column. The PCX columns were activated with 2 mL of methanol before use. The collected extract (0.25 mL) was filtered into a 1.5 mL brown chromatography vial through a 0.22 µm pore size membrane filter. The samples in the bottles were stored in the refrigerator at 4°C under dark conditions awaiting HPLC detection.”. (See in L 90-97).

Question # 4: A standard curve is mentioned for ergonovine but no source or purity of the standard is given.

Answer: According to your suggestions, we have added source or purity of the standard. As following: Ergonovine was purchased from Sigma-Aldrich China. The 0 mg/L, 0.375 mg/L, 0.75 mg/L, 1.5 mg/L, 3 mg/L, 6 mg/L and 12 mg/L concentrations of ergometrine standard solutions were configured by diluting the mother liquor of the standard (20 mg/L). The relevant standard linear equations were established by the external standard method: $Y_{\text{Ergonovine}} = 154.98X - 10.152$, $R^2 = 0.9981$, Y: peak areas, X: Concentration (mg/L). The process was monitored and the peak area was determined using the Chem Station for LC Rev.A.10.01,USA. Alkaloid levels in injection volume was identified and quantified by comparing their peak areas with external standard curves. Ergonovine contents in plant samples was calculated according to the relevant equation and the dilution of the sample. (See in L 105-110)

Question # 5: For the microbiome analysis the primers that were used are not suitable for plant studies as they are well known to detect large amounts of non-target DNA (e.g. plant chloroplast and mitochondrial DNA). It looks like this mistake was made as the primers were taken from a paper (Liu et al) that investigated bacteria from waters. Therefore, Liu et al did not extract microbial DNA from plants. It is possible to use blockers but the authors never mentioned this. I instead have suggested the use of alternative primers that are more suitable for amplification of DNA from plant material.

Answer: Previous studies have found that subsequent analysis of plant endophyte bacterial data were indeed interfered due to the generation of large amounts of plant-derived DNA (plant mitochondrial 16SrDNA and chloroplast 16SrDNA) contamination in the amplicon libraries. Because plant mitochondrial 16SrDNA (Mt16S), chloroplast 16SrDNA (Ct16S) and bacterial 16S rDNA exhibit a certain percentage of sequence homology over their full lengths. To avoid interference of plant chloroplast and chloroplast DNA, amplification of the 16S rRNA gene was carried out using primers primers 799F/1193R, F27/R1530 and nested PCR.

However, we chose the primer (338F: 5'-ACTCCTACGGGAGGCAGCAG-3' and 806R: 5'-GGACTACHVGGGTWTCTAAT-3') based on three considerations in our study. As following:

1. Our initial experimental conception was to compare the differences between shoots, roots and soil microbes, so primers 338-806 were used in order to detect as many microorganisms as possible.
2. Our experiment wanted to determine the effect of *Epichloë* endophyte infection on plant microorganisms. Two populations of *Achnatherum inebrians* seeds that differed only in the presence or absence of the *Epichloë* endophyte were compared on microbial differences in shoot and root. Therefore, we suggest that endophyte-infected plants and endophyte-free plants share the same plant mitochondrial and chloroplast DNA information, and our sequencing results yielded the same results.
3. Plant endophyte primers were designed to avoid interference caused by plant host information, but more host information was not found during our measurement. Meanwhile, in subsequent sequence quality analysis, the sequences from plant chloroplast and mitochondria was filtered. Both the plant endophyte primer and this primer were able to reflect the characteristics of the microbial community within the micro-environment. I will carefully analyze which primers to use in future sequencing experiments.

Question # 6: The UHPLC analysis also refers to citations that are not suitable. For instance, the Gonzalez et al 2011 paper is cited for a protocol but this citation is concerned with UHPLC of rat brain samples. The

citation contains no protocol matching the current paper and does not contain any instrument settings.

Answer: Revised: Thanks for your comments. According to your suggestions, we have changed “González RR, Fernández RF, Vidal JLM, Frenich AG, Pérezmg. 2011. Development and validation of an ultra-high performance liquid chromatography–tandem mass-spectrometry (UHPLC–MS/MS) method for the simultaneous determination of neurotransmitters in rat brain samples. *J Neurosc Meth* 198:187-194. <https://doi.org/10.1016/j.jneumeth.2011.03.023>.” into “Li C, AI-Dalali S, Zhou H, Xu BC. 2022. Influence of curing on the metabolite profile of water-boiled salted duck. *Food Chem* 397:133752-133767. <https://doi.org/10.1016/j.foodchem.2022.133752>.”. (See in L 502-503).

Question # 7: Please see the attached WORD document for a more detailed review.

Answer: We found no evidence of reviewer revisions in the attached WORD document, but we made detailed revisions throughout the text (See in Manuscript-Jinyy-modified original version). As following:

ABSTRACT

The clavicipitaceous fungus *Epichloë gansuensis* forms symbiotic associations with drunken horse grass (*Achnatherum inebrians*), providing biotic and abiotic stress protection to its host. However, it is unclear how *E. gansuensis* affects assembly of host plant-associated bacterial communities after ammonium nitrogen ($\text{NH}_4^+\text{-N}$) treatment. We examined the shoot- and root-associated bacterial microbiota, and root metabolites of *A. inebrians* when infected (I) or uninfected (F) with *E. gansuensis* endophyte. The results showed more pronounced $\text{NH}_4^+\text{-N}$ -induced microbial and metabolic changes in the endophyte-infected plants compared to the endophyte-free plants. *E. gansuensis* significantly altered bacterial community composition and β -diversity in shoots and roots, and increased bacterial α -diversity under $\text{NH}_4^+\text{-N}$ treatment. The relative abundance of 117 and 157 root metabolites significantly changed with *E. gansuensis* infection under water and $\text{NH}_4^+\text{-N}$ treatment compared to endophyte-free plants. Root bacterial community composition was significantly related to the abundance of the top 30 metabolites (VIP>2 and VIP>3) contributing to differences between I and F plants, especially alkaloids. The correlation network between root microbiome and metabolites was complex. Microorganisms in the Proteobacteria and Firmicutes phyla were significantly associated with the R00693 metabolic reaction of cysteine and methionine metabolism. Co-metabolism network analysis revealed common metabolites between host plants and microorganisms. (See in L 18-30).

IMPORTANCE

Our results suggest that the effect of endophyte infection is sensitive to nitrogen availability. Endophyte symbiosis altered the composition of shoot and root bacterial communities, increasing bacterial diversity. There was also a change in the class and relative abundance of metabolites. We found a complex co-occurrence network between root microorganisms and metabolites, with some metabolites shared between the host plant and its microbiome. The precise ecological function of the metabolites produced in response to endophyte infection remains unknown. However, some of these compounds may facilitate plant-microbe symbiosis by increasing the uptake of beneficial soil bacteria into plant tissues. Overall, these findings advance our understanding of the interactions between the microbiome, metabolome, and endophyte symbiosis in grasses. The results provide critical insight into the mechanisms by which the plant microbiome responds to nutrient stress in the presence of fungal endophytes. (See in L 33-40).

INTRODUCTION

The perennial grass *Achnatherum inebrians* is highly adaptable, allowing it to dominate degraded grasslands across northwest China from 1,700-4,300 meters elevation. This 30-million-hectare range makes it an important species for controlling soil erosion and stabilizing sandy soils (1). However, its common symbiosis with the fungal endophytes *Epichloë gansuensis* and *E. inebrians* poses problems for livestock. These endophytes increase the plant's resilience, productivity and adaptability but produce toxic alkaloids (ergine and ergonovine) that can poison grazing animals (2, 3). Replacing infected grasses with endophyte-free cultivars is impractical because the fungi provide ecological benefits (4, 5). (See in L 45-50).

The advantages of the *Epichloë* symbiosis are well-studied. The fungi enhance the grass's drought and

cold tolerance (6, 7), growth (8, 9), nutrient uptake (10-12), disease and pest resistance (13-16), tolerance of heavy metals (17, 18), and protecting biodiversity of over-grazed grasslands (19). They induce physiological and morphological changes in the host such as altered root-shoot ratios (8), increased photosynthesis (9), and shifts in sugars, amino acids, and enzymes (18). While the benefits of *Epichloë* endophyte to host health and growth are clear, the influence on above-ground and below-ground microorganisms is less understood. (See in L 51-56).

E. gansuensis infection altered the composition and diversity of bacterial and fungal community in phyllosphere and seed of *A. inebrians* in the field (20, 21), but there have been surprisingly few the effects of environmental components interacts with *Epichloë* endophyte on plant microbiome. Comparative studies reveal endophyte-related differences in root soil archaea and bacteria involved in nitrogen cycling (22). The fungi also increase root fungal diversity under drought and alter soil bacterial communities (23-25). However, the mechanisms behind these microbiome shifts are unknown. (See in L 57-61).

The symbiosis alters host metabolism by changing pathways and alkaloid production (26, 27). These changes affect litter decomposition and soil microflora (28-30). Foliar endophytes likely influence root microbes through both litter chemistry and root exudates (31). Nevertheless, we lack a comprehensive understanding of the specialized metabolites produced, their functions, and their microbial effects (32, 33). Among the unknowns are the dynamics of microorganisms from host shoot and root tissues during growth, the metabolic processes that are engaged during microbial changes, and regulatory relationships between root microbes and root exudates. Knowledge of the latter is important if we hope to clarify the effect of foliar endophytes on leaf, root and soil microorganisms in drunken horse grass via integrating microbiology and metabolomics analysis. (See in L62-68).

This knowledge gap motivates the current study. We will monitor changes in shoot and root microbes and exudates in *A. inebrians* with and without *E. gansuensis* under different nitrogen regimes. The goals are to identify responsive microorganisms and metabolites and elucidate how endophyte symbiosis affects plant-microbe interactions and specialized metabolism under varying nutrient availability. Overall, this introduction summarizes the current state of knowledge, identifies key gaps, and outlines an approach to advance our understanding of the complex interplay between the microbiome, metabolome, and symbiosis in this ecologically important grass. (See in L69-74).

MATERIALS AND METHODS

Plant material and treatment

Epichloë gansuensis endophyte-infected (I) and -free (F) *Achnatherum inebrians* seeds were collected from E+ and E- seed fields, and planted at Yuzhong Experimental Station of Lanzhou University, Gansu Province, China, in 2017. All the seeds were stored at 4°C in the dark until the following experiments. I and F seeds were planted in 72 holes plate of cultivation seedlings filled with vermiculite (2.0 kg), which was sterilized at 150°C for 24 h. After 45 days, endophyte status detection in seedlings was performed by staining and microscopic examination following the method of Welty (34). All the I seedlings infected with endophyte and then 8 seedlings that differed only in the presence or absence of the *Epichloë* endophyte were selected for the following experiments, respectively. Endophyte-infected seedlings and endophyte-free seedlings were transferred into water (W) and 0.1 mmol/L ammonia-N (A) nutrient solutions. Plants were grown in a greenhouse at 25 ± 2°C with a 16 h: 8 h, light: dark photoperiod (c. 600 μmol.m⁻². s⁻¹) in a random block design and at 8 wk post-cultivation. Root exudates were collected using a root exudates collection device after plants reached 7 days of growth. Leaf and root tissues of I and F plants (n = 4 per group) under different nutrient treatments were collected after 9 wk (8 wk+7 days) at a consistent time and immediately flash-frozen in liquid nitrogen (N₂). (See in L76-88).

Ergonvine extraction and determination

Alkaloid was extracted in two stages using a two-phase solvent system following a method modified from Miles et al. (35). In brief, the seedlings were frozen in the refrigerator at -20°C for 5 h and then dried in a freeze dryer at -60°C for 24 h. The freeze-dried material was ground to a fine powder with a mortar. A 50 mg powder sample was taken into 2 mL centrifuge tube, and 1 mL 20% ice acetic acid was added, followed

by ultrasonication for 5 min, shaking for 2 min, and centrifugation for 5 min at 1000 r/min. The alkaloids extract was collected with 1 mL miscible liquids of 95% methanol and 5% ammonia after purification by filtration through a PCX column. The PCX columns were activated with 2 mL of methanol before use. The collected extract (0.25 mL) was filtered into a 1.5 mL brown chromatography vial through a 0.22 µm pore size membrane filter. The samples in the bottles were stored in the refrigerator at 4°C under dark conditions awaiting HPLC detection. (See in L90-97).

Ergonovine was quantified on an Agilent 1100 high performance liquid chromatography system (HPLC), with a 250 mm×4.6 mm Eclipse XDB-C18 column containing 5 µm particles. Mobile phase: 0.1M NH₄OAc (A): 0.1M CH₃CN=3:1, at a flow rate of 1.0 mL/min, 5 µl injection volume, and open the wash needle (wash with 10% isodiol). The eluent gradients: 0.0 min-10.0 min: 95% A: 5% B; 10.0 min-25.0 min: 75% A: 25% B; 35.0 min-50 min: 50% A: 50% B; 51.0 min-55.0 min: 5% A: 95% B; 56.0 min-71.0 min: 95% A: 5% B. The detection wavelength (Ex) of a fluorescence detector was set at 312 nm, the emission wave (Em) was 427 nm, and the column temperature was 25°C. Ergonovine was purchased from Sigma-Aldrich China. The 0 mg/L, 0.375 mg/L, 0.75 mg/L, 1.5 mg/L, 3 mg/L, 6 mg/L and 12 mg/L concentrations of ergometrine standard solutions were configured by diluting the mother liquor of the standard (20 mg/L). The relevant standard linear equations were established by the external standard method: $Y_{\text{Ergonovine}} = 154.98X - 10.152$, $R^2 = 0.9981$, Y: peak areas, X: Concentration (mg/L). The process was monitored and the peak area was determined using the Chem Station for LC Rev.A.10.01, USA. Alkaloid levels in injection volume was identified and quantified by comparing their peak areas with external standard curves. Ergonovine contents in plant samples was calculated according to the relevant equation and the dilution of the sample. (See in L98-110).

RESULTS

Differential assemblage of shoot and root-associated bacterial microbiota in response to the presence or absence of *Epichloë* endophyte

The findings of community composition analysis revealed that N addition had the greatest significant impact on the change of the plant microbiome (Fig S1 and Fig 1). Further, it was found that shoots microbiomes in shoots of endophyte-infected plants consisted mostly of Cyanobacteria, Proteobacteria, Bacteroidota and Actinobacteriota under nitrogen treatment, accounting for >90% of total relative abundance, with Cyanobacteria dominating the shoots-associated bacteria of endophyte-free plants (Fig 1 a). The root-associated bacterial communities of endophyte-infected plants, at the phylum level, were mostly dominated by Proteobacteria (41.71%), Bacteroidota (37.62%), Cyanobacteria (9.16%) and Actinobacteriota (5.64%), but for endophyte-free plants, Firmicutes (65.34%) and Proteobacteria (24.71%) was the most dominant phylum (Fig 1 b). It was observed that the *E. gansuensis* infection significantly increased the relative abundance of Proteobacteria in roots from 24.63% to 42.18% relative to endophyte-free plants, Bacteroidota from 4.72% to 37.47%, and Actinobacteriota from 1.27% to 5.63% (Fig 1 c to f). The *rrn* copy number of rare communities were significantly changed by the presence of the endophyte under A-N treatment (Fig S2). (See in L204-215).

(Fig 1)

The Shannon index and Chao 1 index of bacterial communities in shoots and roots were significantly ($P < 0.05$) influenced by the presence of the *Epichloë* endophyte under N treatment, but showed no significant difference under water treatment (Fig 2 a to d). Meanwhile, principal co-ordinates analysis (PCoA) was used to visualize the β-diversity based on the Bray-Curtis dissimilarity, and ANOSIM analyses were conducted to evaluate the significance of changes in community structure ($P < 0.05$) between endophyte-infected plants and endophyte-free plants (Fig 2 e to h). (See in L217-221).

(Fig 2)

Achnatherum inebrians-*Epichloë gansuense* symbiont show N-inducible root metabolite alterations

We identified 3003 and 2991 mass spectrometry peaks in positive and negative ion mode (QC) samples (Table S1). In total, 604 and 207 metabolites were successfully detected from I and F roots of *A. inebrians* in

POS and NGE modes before data preprocessing, respectively. To eliminate or reduce the error caused by the analysis process, 580 and 198 POS and NEG modes differential metabolites with names features were selected for downstream statistical analysis after filtering of low-quality peaks, missing value padding, data normalization and RSD assessment of quality control (Table S1). Nitrogen treatment altered the ratio of the down-regulated and the up-regulated metabolites in root exudates in POS and NGE modes between endophyte-infected plants and endophyte-free plants (Fig 3 a, b and Fig S3). The OPLS-DA models results showed that these detected metabolites were significantly diverse between I and F root under A-N ($R^2X = 0.564$, $R^2Y = 0.992$, $Q^2Y = 0.855$) and water ($R^2X = 0.858$, $R^2Y = 0.999$, $Q^2Y = 0.924$) treatments (Fig 3 c and d). (See in L223-232).

(Fig 3)

Aligned with the microbiomic changes, the analysis revealed that *E. gansuense*-infected *A. inebrians* had a greater number of specific metabolites than *E. gansuense*-free *A. inebrians* under N treatment (Fig 4 b). These differential metabolites could be divided into 9 classes, including organic acids and derivatives, lipids and lipid-like molecules, organo-heterocyclic compounds, organic oxygen compounds, alkaloids and derivatives, and Benzenoids (SUPPLEMENTAL FILE 2 and Table S2). Our analysis of the top 30 metabolites ($VIP > 2$ and $VIP > 3$) contributing to the difference between I and F plants revealed that most compounds significantly increased in abundance by endophyte infection under A-N treatment (Fig 4 c), whereas a significant decrease was observed under water treatment compared to endophyte-free plants root (Fig 4 d). Among the up-annotated signature metabolites detected under water treatment, general classification metabolites were highly enriched (Fig 4 c). In contrast, except for the significant enrichment of general metabolites, some specific metabolites were also significantly enriched in the root of endophyte-infected plants root under A-N treatment (Fig 4 d and SUPPLEMENTAL FILE 2). The specific metabolites identified by human metabolome database (HMDB) searches as 6-allyl-8b-Carboxy-ergoline, Cinchonidine, Norajmaline and (2-Naphthalenyloxy) acetic acid were significantly accumulated in roots of endophyte-infected plants compared to endophyte-free plants (abundance all zero) under A-N treatment (Fig 4 e, f, h and i). In addition to ergometrine were significantly increased by 732-fold in roots of endophyte-infected plants than roots of endophyte-free plants (Fig 4g), the results of *t*-test showed that ergonovine content in shoot under water ($P=0.005$) and A-N ($P<0.001$) treatments was significantly increased by endophyte infection (Fig.S4). (See in L234-249).

(Fig.4)

There were 6 metabolite pathways significantly enriched ($P<0.05$) in *A. inebrians* under water treatment (Fig S5 a). Compared to water treatment, the significantly enriched pathways for 157 metabolic compounds in I and F plants by the hypergeometric distribution algorithm and relative-betweenness centrality were phenylalanine metabolism ($P=0.002$), aminoacyl-tRNA biosynthesis ($P=0.030$), lysine degradation ($P=0.025$), tryptophan metabolism ($P=0.035$) and cysteine and methionine metabolism ($P=0.036$) under A-N treatment (Fig S5 b). (See in L251-255).

Visualization of biological and statistical correlation between bacterial community and root exudates using Sankey network

According to the metabolites from microbiota, three significant co-metabolism associated metabolic pathways were identified associated with nutrient supply, including cysteine and methionine metabolism, tyrosine metabolism, and arginine and proline metabolism. For instance, five differential metabolites (i.e., S-Adenosylmethioninamine, spermidine synthase, adenosylmethionine decarboxylase, S-Adenosyl-L-methionine and 5'-Methylthioadenosine) were involved in the cysteine and methionine metabolic pathway, which participated in three different metabolic reactions (R00178, R01920 and R02869). Proteobacteria, Firmicutes, Actinobacteria and Bacteroidetes phyla were closely associated with metabolic reaction R00693 in the Bio-Sankey network (Fig.5 a). The relative abundance of S-Adenosylmethioninamine in root metabolism was significantly up-regulated in endophyte-infected plants than endophyte-free plants under A-N treatment (SUPPLEMENTAL FILE 2). The relative abundance of *Bacillus*, *Paenibacillus* and *Exiguobacterium* genera in roots was the most significantly increased in *A.*

inebrians plants under A-N treatment (dark red if $P < 0.05$), and was higher in endophyte-free plants compared to endophyte-infected plants, which were closely associated with metabolic reaction R01920 in parallel (Fig 5 a). Statistical correlation analysis confirmed that Proteobacteria phylum were closely associated with S-Adenosylmethioninamine in the metabolic reaction R00693 by the STA-Sankey network (Fig 5 b). (See in L256-270).

(Fig5)

Co-Metabolism network shows differential metabolites shared by the host and microbiota, and their related microbes

The co-metabolism network of cysteine and methionine metabolism, tyrosine metabolism, and arginine and proline metabolism showed that two root metabolites were associated with 49 differential root bacteria under A-N treatment ($P < 0.05$) (Fig 6). With the exception of very few bacteria, for example, *Enterococcus*, *Exiguobacterium*, *Paenibacillus* and *Fluviicola* genera, most of bacterial communities were significantly and positively correlated with S-Adenosylmethioninamine. However, we were able to find a significant positive effect of L-Dopa metabolites only on bacteria of the genus *Bacillus*, but significant negative correlation with other bacterial community (Fig 6). Bacteria-root metabolite association network provided more specific information on metabolic changes associated with *A. inebrians*-*E. gansuense* symbiont plants under A-N treatment. (See in L272-281).

(Fig 6)

DISCUSSION

By performing biological and statistical correlation analyses between root bacteria and metabolites, this study provides novel evidence for the numerous close and intricate interrelationships between the two factors. The metabolomic data showed the *E. gansuensis* endophyte altered metabolic components in host plants. Integration of microbiome and metabolome data sheds light on the complex bidirectional interactions between endophytes, host plants, and microbial communities. Our multi-omic characterization of endophyte-infected *A. inebrians* advances understanding of the intricate network of signals mediating tripartite symbiosis. Further elucidation of the specific mechanisms underlying microbiome-metabolome interplay will be key. Overall, this systems biology approach coupling amplicon sequencing and metabolomics enables greater insight into the multidimensional impacts of endophytes on host-microbe interactions. (See in L283-290).

Effect of the *E. gansuensis* endophyte on the composition of plant bacterial microbial communities

The current study found that *A. inebrians* plants infected with the *E. gansuensis* endophyte harbored more complex phyllosphere microbial communities compared to uninfected plants (Fig 1a and 1b). These results align with previous studies by Liu et al. and Liang et al. showing altered microbiome composition and diversity in the phyllosphere, seeds, and roots of endophyte-infected *A. inebrians* under optimal conditions (20, 21). Extensive research on other *Epichloë* endophytes like *E. occultans*, *E. bromicola*, and *E. festucae* var. *lolii* has also shown that root and rhizosphere microbiomes exhibit complex responses to endophyte symbiosis (46-48). Our findings further demonstrate that shifts in plant microbial communities can be driven by the presence of fungal endophytes (22, 25), reflecting the critical role endophytes play in host adaptation. (See in L292-298).

The current study demonstrated that the A-N treatment induced significant alterations in bacterial community composition in both shoots and roots of endophyte-infected plants compared to endophyte-free plants (Fig 1a and 1b). This manifested as a greater fold-change increase in bacterial abundance in infected versus uninfected plants under A-N conditions (Fig 1e and 1f). These findings illustrate an amplified nitrogen-responsive effect of the endophyte, aligning with previous research showing the influence of *Epichloë* endophytes on host plants is enhanced in nitrogen-rich environments (49). As proposed by prior studies, copiotrophic bacteria thrive in nutrient-rich conditions while oligotrophic bacteria with efficient nutrient utilization prevail in nutrient-poor settings (50). Accordingly, we found endophyte-infected shoots harbored more rare bacteria compared to endophyte-free shoots under A-N, while the opposite trend occurred in roots, mirroring measured nitrate and ammonium levels (Fig. S6). As rare microbes play key

roles governing nitrogen availability and plant nutrient uptake (51), the endophyte-mediated shifts in low abundance bacteria may have significant ecological impacts. (See in L299-308).

Numerous studies demonstrate shoot and root microbiome alpha diversity critically regulates plant health, growth, and environmental adaptability (52, 53). Our work revealed the *Epichloë* endophyte markedly increased alpha and beta diversity of shoot and root bacteria under A-N but not under water-only conditions (Fig 2), aligning with previous findings (20, 21, 54). Taken together, these results suggest nitrogen availability enables endophytes to substantially augment the diversity and complexity of host-associated bacterial communities. Further investigation of the multifaceted interplay between nitrogen, endophytes, and microbiota will provide greater insight into the factors shaping holobiont resilience. Our integrative analysis of amplicon data with plant metabolomic and nutrient profiles represents a valuable systems biology approach to elucidating the intricate tripartite network of endophyte-host-microbe interactions. (See in L309-316).

Effect of the *E. gansuensis* endophyte on root metabolites

This study demonstrated that *E. gansuensis* infection was the primary driver of metabolic profile alterations in *A. inebrians*, as endophyte-infected and endophyte-free plants showed distinct profiles even under optimal conditions (21). Notably, we observed amplified nitrogen-responsive metabolic divergence between infected and uninfected plants (Fig 3), aligning with previous findings on low-nitrogen responses (49). General and specialized metabolism pathways exhibited pronounced differences between endophyte-infected and endophyte-free roots under A-N (Fig 4), mirroring the accumulation of various organic acids, amino acids, and fatty acids reported by Hou et al. (49). Other metabolomic studies reveal *Epichloë* endophytes can modulate host stress resilience, growth, and soil insect distributions by altering root exudate composition and metabolic pathways (55, 56). (See in L318-324).

We found *A. inebrians* alkaloid contents increased under salt and drought stresses, particularly cytotoxic ergonovine (57). Similarly, under A-N conditions, endophyte-infected plants showed significant ergometrine accumulation in root exudates (Fig S4) -contrasting related stress-induced alkaloids like ergine and ergonovine (49, 58). The major contribution of toxic pyrrolopyrazine and ergot alkaloids to nitrogen uptake and translocation in infected plant shoots may relate to the concurrent accumulation of carbohydrates and amino acids, as seen in endophyte-infected tall fescue (59). (See in L325-329).

Overall, our dual metabolomic and amplicon sequencing analyses provide novel insight into the close interrelationship between the endophyte-conferred alkaloid profile and the structure of the root microbiome. The significant correlations found between specific bacterial taxa and ergot alkaloids hint at a complex codependency. Further elucidation of the factors regulating alkaloid biosynthesis and secretion, the bioactivities of these compounds against different microbes, and subsequent microbial community shifts will be key. This holistic multi-omic approach enables systems-level understanding of the multifaceted chemical signals mediating dynamic endophyte-host-microbe interactions belowground. (See in L330335).

Root microbial and metabolic interactions with *Achnatherum inebrians*-*Epichloë gansuense* symbiont

Prior studies show root exudates are the primary nutrient source for rhizosphere microbes, directly shaping ecological and compositional dynamics (60). Notably, correlation analysis revealed significant positive associations between root metabolite levels and bacterial taxa, suggesting endophyte infection may recruit communities by modulating metabolites (Fig.S7). Microbes chemotactically locate root tips using exuded sugars, organic acids, and amino acids but are then degraded by reactive oxygen species as the plant extracts nutrients from colonizers (61, 62). We found the antioxidant glutamylcysteine was markedly accumulated in endophyte-infected roots under A-N, potentially mitigating reactive oxygen and improving nutrient extraction from bacteria. (See in L336-343).

The co-occurrence network demonstrated significant interrelationships between the root microbiome and metabolome in *A. inebrians* (Fig 5). For instance, we observed nitrogen-induced accumulation of S-adenosylmethioninamine exclusively in endophyte-infected roots, highlighting metabolite co-metabolism between bacteria and the symbiotic holobiont (44). (See in L344-346).

Overall, integration of amplicon and metabolomic data provides greater insight into the chemical dialog regulating tripartite interactions belowground. Our findings reveal endophytes modify root exudate profiles to selectively enhance colonization by beneficial microbiota, highlighting the key role of plant-microbe signaling in symbiotic assembly and function. Further characterization of metabolite bioactivities, specialized microbial nutritive requirements, chemoattraction dynamics, and community interdependencies will elucidate the multifaceted mechanisms enabling robust, mutually beneficial endophyte-host-bacteria relationships.(See in L347-352).

Conclusions

Nitrogen significantly amplified the effects of the *Epichloë* endophyte on host-associated microbes and metabolites compared to water-only conditions. Our results demonstrate the *E. gansuense* endophyte induced substantial alterations in shoot and root bacterial community composition and beta diversity under NH_4^+ -N treatment. Relative abundance of rare taxa in shoots and alpha diversity in both shoots and roots were higher in infected versus uninfected plants under nitrogen fertilization. Concurrently, endophyte infection modulated root metabolite classes and abundances, with variations in alkaloid levels linked to shifts in the root microbiome assembly. Complex and significant correlations were found between root bacteria and metabolites, highlighting their multifaceted interplay. This integrated amplicon sequencing and metabolomics approach provides novel insight into the specialized metabolites secreted by endophyte-infected plants and their relevance in recruiting beneficial microbiota. Our findings exemplify the power of combinatorial multi-omics to elucidate the intricate chemical dialog mediating tripartite endophyte-host-microbe interactions belowground. Further characterization of metabolite bioactivities, chemoattraction mechanisms, and microbial dependencies will be key to deciphering the complex signaling enabling robust, mutually beneficial symbioses. Overall, this systems biology strategy represents a valuable tool for rapidly advancing discovery of bioactive compounds and clarifying their ecological roles.(See in L355-367).

Revision Notes-Response to Reviewer 2 Comments:

Dear the reviewers 2:

Thank you for your approval and constructive suggestions concerning our manuscript entitled "Interactions between *Epichloë* endophyte and the plant microbiome impact nitrogen responses in host *Achnatherum inebrians* plants (Spectrum02574-23)", which we submitted to Microbiology Spectrum. Those comments are all valuable in improving our manuscript. We have studied comments carefully and made revisions accordingly which we hope it meets with your approval.

Point-by-point responses to the issues raised by the reviewer 2:

Question # 1: Does the author only focus on the host metabolites ? or the microbial metabolites that are present inside the plants? During treatment not only the host but microbial metabolites are also produced in different response.

Answer: We measured root metabolites of host plants, including host metabolites and endophyte microbial metabolites in root. Our main objective was to compare the differences on root exudates between endophyte-infected plants and endophyte-free plants, so we did not make a specific distinction between plant metabolites and microbial metabolites.

Question # 2: Line 93. Delete this line "The results of this work are presented here."

Answer: According to your suggestions, we have deleted this line "The results of this work are presented here.". (See in L187).

Question # 3: Please add data of culturable endophytes if you have after you treated the host plant and check the microbial diversity.

Answer: In this study, we investigated the effects of *Epichloë* endophyte on the composition and diversity of bacterial microbial communities in shoots and roots of host plants under water and nitrogen treatments. However, we did not isolate and characterize bacteria from leaves and roots of host plant. According to your suggestions, we will consider the isolation and identification of culturable endophytes from host plants in our subsequent studies.

Question # 4: How about quantification of metabolites?

Answer: The relative abundance of metabolites were measured by the UHPLC-MS/MS analysis in our manuscripts, and absolute quantification of metabolites was not measured.

Re: Spectrum02574-23R1 (**Interactions between *Epichloë* endophyte and the plant microbiome impact nitrogen responses in host *Achnatherum inebrians* plants**)

Dear Prof. Chunjie Li:

Your manuscript has been accepted, and I am forwarding it to the ASM production staff for publication. Your paper will first be checked to make sure all elements meet the technical requirements. ASM staff will contact you if anything needs to be revised before copyediting and production can begin. Otherwise, you will be notified when your proofs are ready to be viewed.

Sincerely,
Victor Gonzalez
Editor
Microbiology Spectrum